# Microwave-transparent metallic metamaterials for autonomous driving safety

Eun-Joo Lee [1], Jun-Young Kim [1], Young-Bin Kim [1] & Sun-Kyung Kim [1] ✉

Maintaining the surface transparency of protective covers using transparent heaters in extreme weather is imperative for enhancing safety in autonomous driving. However, achieving both high transmittance and low sheet resistance, two key performance indicators for transparent heaters, is inherently challenging. Here, inspired by metamaterial design, we report microwave-transparent, low-sheet-resistance heaters for automotive radars. Ultrathin (approximately one ten-thousandth of the wavelength), electrically connected metamaterials on a millimetre-thick dielectric cover provide near-unity transmission at specific frequencies within the W band (75–110 GHz), despite their metal filling ratio exceeding 70 %. These metamaterials yield the desired phase delay to adjust Fabry–Perot resonance at each target frequency. Fabricated microwave-transparent heaters exhibit exceptionally low sheet resistance (0.41 ohm/sq), thereby heating the dielectric cover above 180 °C at a nominal bias of 3 V. Defrosting tests demonstrate their thermal capability to swiftly remove thin ice layers in sub-zero temperatures.

Microwave metamaterials offer a versatile approach to engineering material dispersions, enabling precise control over the transmission, reflection and absorption of microwaves tailored to specific applications[1–3]. The exceptional absorption characteristics of microwave metamaterials with deep-subwavelength profiles have garnered significant attention, thereby creating electromagnetic shielding for electronic devices[4] and living organisms[5], stealth technology for military purposes[6] and frequency filters for wireless communication systems[7]. Despite the extensive exploration of metamaterials in previous studies, research on transmissive metamaterials in the microwave regime remains relatively limited[8], often constrained by low metal filling ratios[9] and their potential applications are yet to be clarified.

Autonomous driving requires the integration of diverse sensors responsible for distinct detection ranges, such as LiDARs, radars, cameras and ultrasonic sensors. Radars operating at 76–81 GHz within the W band exhibit remarkable efficacy for long-range object detection, even under adverse weather conditions[10]. Noteworthily, the progressions of autonomous driving and electric vehicles are closely intertwined, as both technologies aim at transforming the transportation landscape toward a more sustainable, efficient and connected future[11]. However, the absence of an internal combustion engine in electric vehicles lowers the temperature of the bonnet, possibly affecting the reliability and functionality of the sensors mounted near the bonnet in cold settings with frost or fog. Therefore, maintaining clear and unobstructed view for sensors is vital for avoiding hazardous situations in various driving scenarios. To address these concerns, manufacturers employ transparent heaters to swiftly remove the aqueous obstacles formed on the covers of the sensors.

Previously developed transparent heaters work well in the visible region and are integrated into protective covers for defrosting and defogging. Researchers have explored various materials for high-performance transparent heaters, such as transparent conductive oxides (TCO)[12], TCO/silver/TCO multilayers[13–15], silver nanowires[16–20] and carbon-based nanomaterials[21,22] encompassing carbon nanotubes, graphene and microstructured metals[23,24]. However, achieving high transmittance and low sheet resistance, the two critical benchmarks of transparent heaters, is challenging owing to the physical correlation between these quantities. Lowering the sheet resistance increases the density of free charge carriers, which adversely affects the

[1]Department of Applied Physics, Kyung Hee University, Gyeonggi-do 17104, Republic of Korea. ✉e-mail: sunkim@khu.ac.kr

transmittance[25]. Furthermore, few attempts have been made to develop transparent heaters that operate in the microwave regime. A feasible solution for microwave-transparent heaters could be a one-dimensional (1D) metallic wire array[26], akin to the wire-grid polarisers used in display applications[27]. This design can perfectly transmit microwaves exclusively for a specific polarisation. However, any alteration in the polarisation of microwaves reflected from the surroundings undermines their detectability[28]. More importantly, the high transmission of a 1D metallic wire array for a specific polarisation vanishes as the metallic filling ratio increases, which is a general trend for hyperbolic metamaterials[29].

In this paper, we propose ultrathin metamaterials comprising an array of electric inductive-capacitive resonators that achieve perfect transmission at specific microwave frequencies within the W band (75–110 GHz) on a millimetre-thick dielectric cover. The metamaterials were designed to exhibit a high metal-filling ratio in each unit cell by adjusting multiple structural variables of the resonators while maintaining the maximum transmittance at each target frequency. The metamaterials were fabricated by depositing a 300 nm-thick copper (Cu) layer on a 0.7 mm-thick glass cover, followed by spectral characterisation using a vector network analyser with frequency extension into the sub-terahertz range. The complex scattering parameters of the ultrathin metamaterials were obtained to account for the extraordinary transmission observed at specific microwave frequencies. To assess the thermal capabilities of the fabricated transparent heaters, defrosting tests were conducted at deep sub-zero temperatures, which highlighted their remarkably low sheet resistances.

## Results

### Concept of microwave-transparent heaters for automotive radars

Automotive radars tasked with long-range detection are installed on the bonnets of vehicles, which are often placed behind the vehicle emblem for concealment (Fig. 1a). In environments with cold temperatures or temperature fluctuations, thin layers of frost or ice accumulate on the radar cover. This can result in the attenuation of both the incoming and outgoing microwave signals caused by reflection and scattering (Fig. 1b). To swiftly remove these aqueous obstructions, employing a heater with a high metal-filling ratio and low sheet resistance is essential. However, the introduction of a heater inevitably causes severe attenuation of the microwave signals.

In radar modules, a dielectric cover with a thickness of the order of a millimetre serves as a resonant cavity for microwave signals. For our experiments, a 0.7 millimetre-thick glass substrate was prepared to emulate a protective cover. The transmission (represented by the $S_{21}$ scattering parameter) spectrum of an uncoated glass cover was obtained across the W band (75–110 GHz) using a vector network analyser with a frequency extension into the sub-terahertz range (Fig. 1c solid line and Methods). The glass cover exhibited its highest transmission at 93.4 GHz, which corresponds to the first-order Fabry–Perot resonance (i.e. standing wave) frequency. This resonance is governed by the standing-wave condition and is expressed as follows:

$$2\frac{2\pi f}{c} n t + \phi_1 + \phi_2 = 2m\pi \tag{1}$$

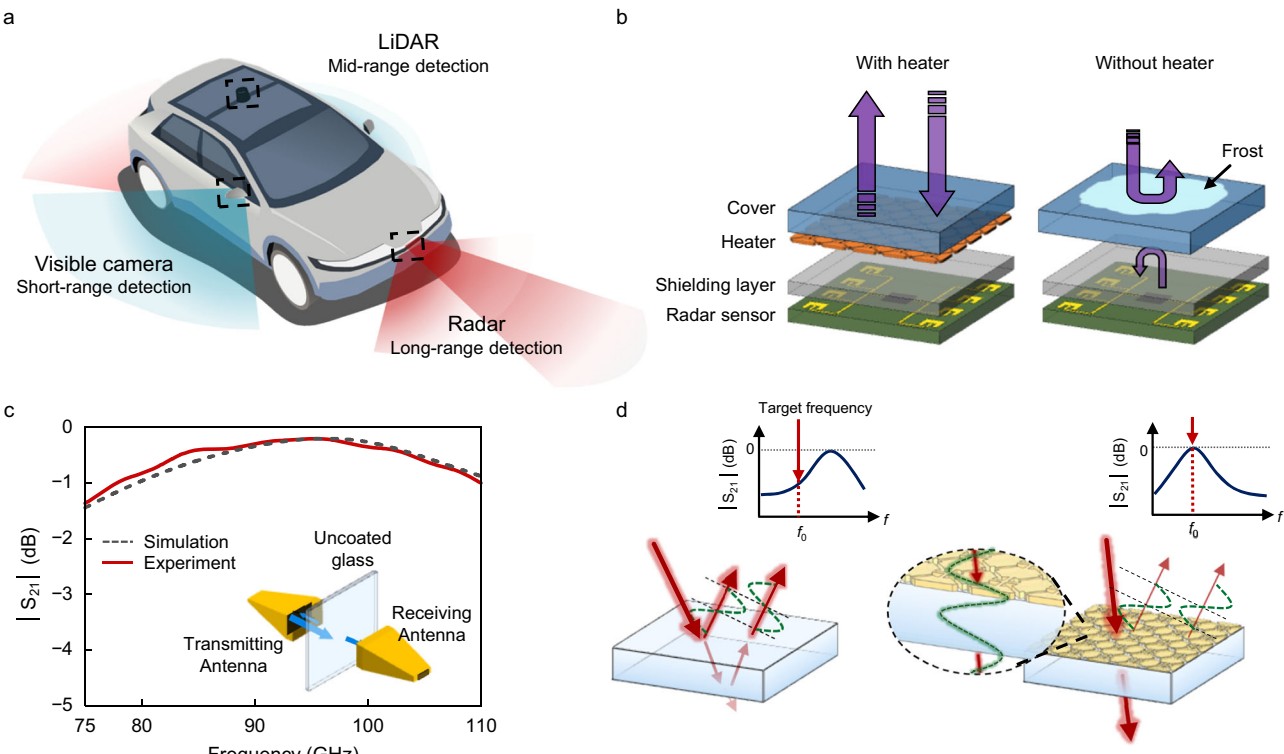

**Fig. 1 | Concept of microwave-transparent heaters for automotive radars.**
**a** Schematic illustrating various types of sensors installed on an autonomous vehicle. **b** Schematics illustrating a radar sensor module with and without a heater in adverse weather conditions. A thin layer of frost or ice accumulated on the cover of the module considerably reflects or scatters both incoming and outgoing signals. **c** Measured and simulated $S_{21}$ spectra of an uncoated glass cover within the W band.

(Inset) Schematic of the measurement setup. **d** Schematics illustrating the principle of a microwave-transparent metamaterial. (Left) An uncoated glass substrate fails to meet the standing wave condition at a target frequency ($f_o$) owing to its inappropriate thickness, resulting in diminished transmission at $f_o$. (Right) A metamaterial inducing an appropriate phase delay adjusts the standing wave condition to $f_o$, enabling perfect transmission at $f_o$.

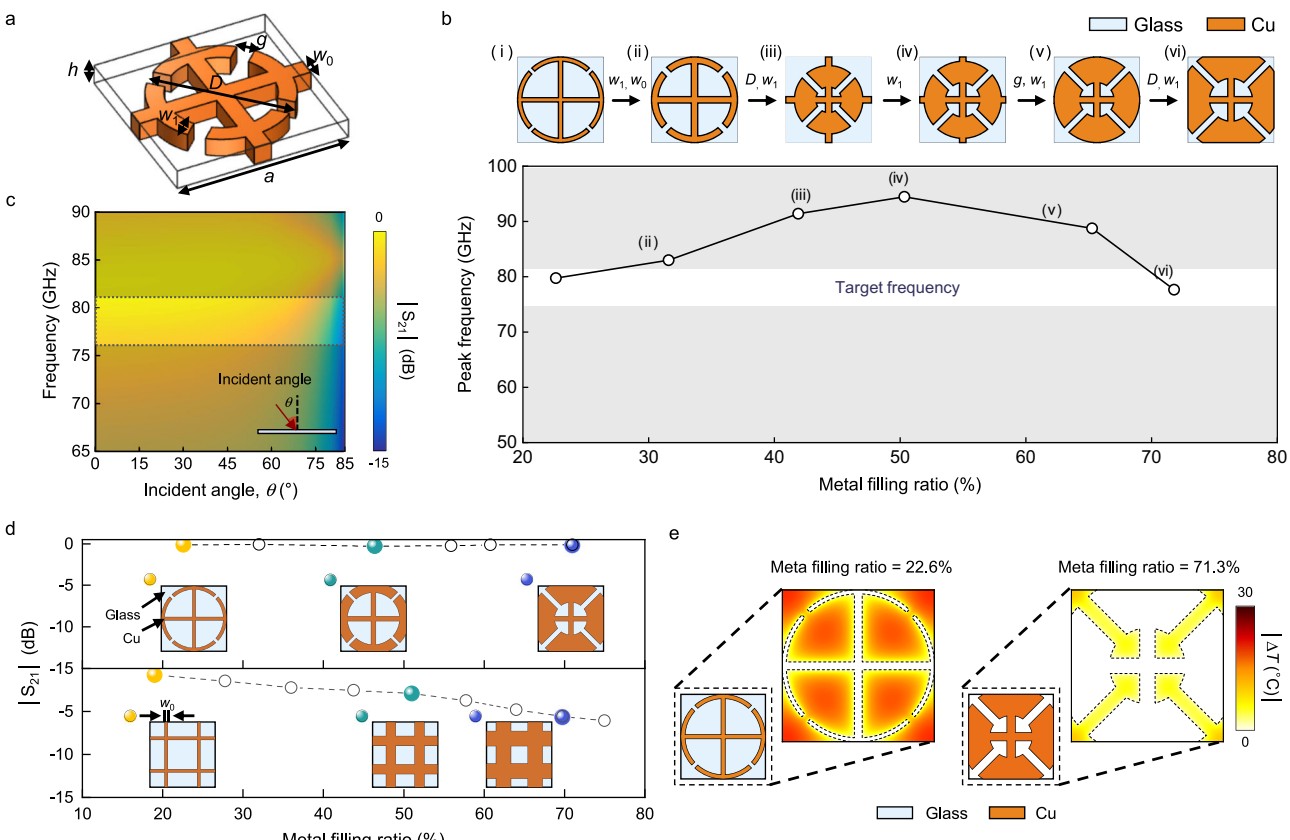

**Fig. 2 | Design of microwave-transparent metamaterials. a** Schematic of a metamaterial design characterised by structural parameters of $D$, $w_0$, $w_1$, $g$, $a$ and $h$. **b** Sequential progression of a metamaterial design, with its metal fill ratio incrementally increasing at each stage. A transmission peak frequency at each stage is plotted. The key parameter values ($D$, $w_0$, $w_1$ and $g$) of each metamaterial design are listed in Supplementary Table 1. For all the simulations, $a = 1000\,\mu m$ and $h = 0.3\,\mu m$. **c** Simulated $S_{21}$ values of the metamaterial design (vi) shown in (**b**) as functions of

incident angle and frequency. **d** Simulated metamaterial designs with various metallic filling ratio. For comparison, two-dimensional mesh arrays with the corresponding metal filing ratios are plotted. The values for structural parameters for the metamaterial and mesh-array are listed in Supplementary Table 2. **e** Temperature distributions on a glass cover incorporating a metamaterial with a low (labelled (i) in **b**) or high (labelled (vi) in **b**) metallic filling ratio, acquired from heat transfer simulations.

where $f$ is the microwave signal frequency, $c$ is the speed of light, $n$ is the refractive index of the substrate and $\phi_1$ ($\phi_2$) is the phase changes when light is reflected from the upper (lower) surfaces of the cover. By inserting the measured thickness and complex permittivity values of the glass cover into the Eq. (1) (Supplementary Fig. 1 and Methods), the standing wave condition was met at 93.4 GHz (Fig. 1c, dashed line). The achieved maximum transmission was slightly short of unity at −0.37 dB (equivalent to approximately 92 %), which was due to the presence of absorption losses within the glass cover. Our strategy involves the design of deep-subwavelength-thick metamaterials that impart an appropriate phase delay to the internally reflected light while retaining its amplitude (Fig. 1d)[30]. These metamaterials adjust the standing wave condition to the desired microwave frequency according to the degree of phase delay, thus enabling perfect transmission at that frequency[31].

**Design of microwave-transparent metamaterials with a high metal-filling ratio**

Our metamaterial design was initiated from a recognised electric inductive-capacitive resonator array[32] to which we added interconnection bars between adjacent unit cells to establish electrical connectivity (Fig. 2a). The design had four primary structural variables ($D$, $w_0$, $w_1$ and $g$) with fixed values of $a = 1000\,\mu m$ and $h = 300\,nm$. In this geometry, increasing the values of $D$, $w_0$ and $w_1$ individually enhanced the metal filling ratio. However, in terms of spectral tuning, an increase in $D$ induced a lower frequency shift of transmission peaks, which is attributed to an increase in the length of inductive loops[33].

Conversely, an increase in $w_0$ and $w_1$ resulted in a peak shift toward higher frequencies, as this effectively reduced the loop inductance (Supplementary Fig. 2 and Supplementary Fig. 3). Although $g$ (the separation between curved units) moderately impacted the metal filling ratio, it significantly tuned the overall spectrum with minimal alteration through the modulation of capacitive coupling. Figure 2b depicts the sequential evolution of the microwave-transparent metamaterial design, with the metal filling ratio progressively increasing at each stage. Through an iterative trial-and-error approach with modifications of the key variables, we identified an initial metamaterial design with a low metal-filling ratio (21.7 %), exhibiting a transmission peak at a specific frequency of 76–81 GHz (labelled (i) in Fig. 2b). By leveraging the spectral tuning characteristics of each variable, we discovered a metamaterial design with the highest metal-filling ratio (71.3 %) (labelled (vi) in Fig. 2b), and its peak frequency matched that of the initial design. The values of the variables for each stage are listed in Supplementary Table 1. Notably, the route of progression is not singular; rather, it can vary depending on the variable values applied to the various sequences. Therefore, designing distinct metamaterials with similar metal filling ratios that exhibit transmission peaks at the same frequency is feasible.

The level of transmission at the peak frequency remained almost unchanged over incident angle of 0–60° (Fig. 2c). Given the numerical aperture of automotive radars, angular tolerance is important[34]. To underscore the extraordinary transmission of the designed metamaterials, we compared their transmission levels with those of two-

dimensional mesh arrays by altering the metal filling ratios in both patterns (Fig. 2d and Supplementary Table 2). The metamaterials were designed to have transmission peaks within the frequency band of 75–80 GHz, despite variations in the metal fill ratios. These metamaterials sustain near-unity ( > −1.0 dB) transmission levels, irrespective of metal filling ratios. In contrast, their counterpart mesh arrays presented a gradual decline in transmission with an increase in the metal filling ratio; the level of transmission was −7.8 dB at the metal filling ratio of 70 %. Heat transfer simulations confirmed that a metamaterial featuring a higher metal-filling ratio guarantees a more uniform temperature distribution (Fig. 2e)[35], which is associated with the effectiveness of the defrosting heaters.

## Spectral and thermal characteristics of fabricated metamaterials

We fabricated large-area $(10 \times 10 \text{ cm}^2)$ microwave-transparent Cu metamaterials on a glass cover using standard semiconductor processes, including photolithography and vacuum deposition (Fig. 3a and Methods). The transmission spectra of the fabricated metamaterial-based samples (labelled Meta I–V) were obtained using a vector network analyser in an anechoic chamber (Fig. 3b solid lines, Supplementary Fig. 4 and Methods). Each metamaterial design possessed a comparable metal filling ratio (67–71 %) while targeting a distinct transmission peak frequency within 75–110 GHz. The values of the primary structural variables for each metamaterial design are listed in Supplementary Table 3. The measured spectra were sinusoidally structured and peaked at 82.5 (Meta I), 87.4 (Meta II), 91.3 (Meta III) and 95.3 GHz (Meta IV). Their maximum transmission levels reached approximately −0.38 dB, identical to that of the uncoated glass cover. This agreement signifies that each metamaterial design led to perfect transmission at its target frequency, even at a high metal-filling ratio. The level of reflection (represented by the $S_{11}$ scattering parameter) was practically zero ( < −25 dB) at each transmission peak frequency (Supplementary Fig. 5a). The measured data were quantitatively supported by electromagnetic simulations using the measured complex permittivity dispersion of the glass cover (Fig. 3b dashed lines).

The uncoated glass cover maintained its transmission peak at 93.4 GHz, as shown in Fig. 1c. However, upon the incorporation of the designed metamaterials into the glass cover, transmission peaks were observed at frequencies other than 93.4 GHz. This observation indicates that each metamaterial design creates an appropriate degree of $\phi_1$ to satisfy the standing-wave condition at the peak frequency. To verify the phase compensation of the metamaterials, we calculated the difference in the phase delay ($\Delta\phi$) that occurred when a microwave signal is reflected from one end of the glass substrate including the metamaterials (Meta I–IV) (Fig. 3c). The simulated $\Delta\phi$ values for the four different metamaterials support the analytical standing wave conditions at 82.2, 87.4, 91.4 and 95.6 GHz, respectively. These frequencies precisely match the transmission peak frequencies observed in the measured $S_{21}$ spectra from Fig. 3b. Despite their deep-subwavelength thickness (approximately 0.1 % of the wavelength), the metallic surfaces effectively induce significant phase changes, behaving as metamaterials with an anomalously high effective permittivity. The simulated results showed that the $\Delta\phi$ values for the employed metamaterial designs (Meta I–IV) exhibited frequency-dependent variations, with their curves shifting equivalently according to the difference in the peak frequency. Notably, the $\Delta\phi$ at the peak frequency led to a standing-wave condition specific to the frequency (Fig. 3d and Supplementary 4b and c). For both measured complex transmission ($S_{21}$) and reflection ($S_{11}$) coefficients, their phase values abruptly shifted from -π to π near the peak frequencies and reached zero at those frequencies. We extracted the electric-field profiles of microwaves traversing the metamaterial (Meta I)-coated glass cover when their frequencies were tuned at 82.5 (on resonance) and 110 GHz (off-resonance) (inset in Fig. 3c). At the resonant frequency, the

incident microwave signal was coupled to the first-order standing wave and consequently, its reflected energy was drastically suppressed.

To facilitate a direct comparison, we fabricated reference samples based on two-dimensional mesh arrays with discrete metal filling ratios (5 %, 30 % and 70 % for Mesh I–III, respectively) and obtained their transmission spectra (inset in Fig. 3e). In contrast to the metamaterial-based samples, the mesh-array-based counterparts exhibited a gradual reduction in transmission with increasing metal filling ratios. At the metal filling ratio of 70 %, the level of transmission was between −13.0 and −7.8 dB within the W band. In addition, we simulated transmittance spectra of 1D metallic wire arrays with different metal filling ratios (10 %, 30 %, 50 % and 70 %) for two orthogonal polarisations (Supplementary Fig. 6). At a metal filing ratio of 70 %, the highest levels of transmission in the W band for transverse magnetic and electric polarisations were −1.9 and −19.7 dB, respectively. The 1D metallic wire arrays exhibited a gradual decrease in transmission when the metal filling ratios increased, as observed in the mesh arrays.

To evaluate the electrical characteristics, we conducted four-point probe measurements on all the fabricated samples (Fig. 3e). The sheet resistance was exclusively governed by the metal filling ratio and was not influenced by the pattern type (metamaterial vs. mesh). Particularly, the samples (Meta I–IV and Mesh III) featuring a metal filling ratio of approximately >65% yielded sheet resistance values of 0.41–0.61 Ω/sq, exhibiting competitive performance compared with transparent heaters operating in the visible regime[15–18,20,23,24,36–44]. In comparison, the sample with a nominal metal-filling ratio of 5 % (Mesh I) exhibited a sheet resistance of >5 Ω/sq. Recent studies reported transparent heaters in the visible achieving sheet resistance values of 0.28 Ω/sq using a 10 μm thick Ag layer[23] and 0.03 Ω/sq from Ag nanoparticle-decorated Cu films with a thickness of 5.1 μm[24]. The sheet resistance of our 300 nm thick, Cu-based metamaterial heaters can be further enhanced by employing more conductive metals (e.g., Ag and Ag-based alloys) and increasing the metal thickness. Although increasing the metal thickness of Mesh I to tens of microns can effectively reduce sheet resistance, this approach degrades transmission within the W band (Supplementary Fig. 7) and leads to nonuniform temperature distribution due to the low metal filling ratio of the mesh array (Supplementary Fig. 8).

## Defrosting tests with microwave-transparent metamaterial heaters

Heating experiments were conducted to assess the capabilities of the fabricated samples as electrical heaters. First, we monitored the temperature changes of the metamaterial-based (Meta III in this instance) sample, while varying the input voltages from 0 to 3 V in increments of 0.5 V (Fig. 4a and Methods). To establish the electrodes, a thermally conductive Cu foil tape was affixed to both edges of the metamaterial heater. For the control experiments, we selected Mesh I, which was characterised by a metal filling ratio of 5 %, for the mesh-array-based sample. This selection was based on a decent level of microwave transmission, which was slightly lower than that of its metamaterial-based counterpart (inset in Fig. 3e). Notably, the thermocouple recorded temperature changes on the uncoated (as opposed to metamaterial-coated) face of the glass cover, replicating practical conditions. According to the measured results, Meta III significantly outperformed Mesh I across all tested input voltages. At 1.5 V, Meta III reached a temperature of 76 °C, which increased to 180 °C at 3 V. In contrast, Mesh I achieved temperatures of 32 °C and 46 °C under the same conditions. The equilibrium temperatures of the samples remained nearly constant for more than 30 min (Supplementary Fig. 9). We evaluated the temperature uniformity across the entire sample area by recording the temperatures at designated positions, including both the centre and edges (Supplementary Fig. 10). In the case of Meta III, when an input voltage of 1.6 V was applied, the measured temperature was 65–80 °C. Notably, the temperature variations

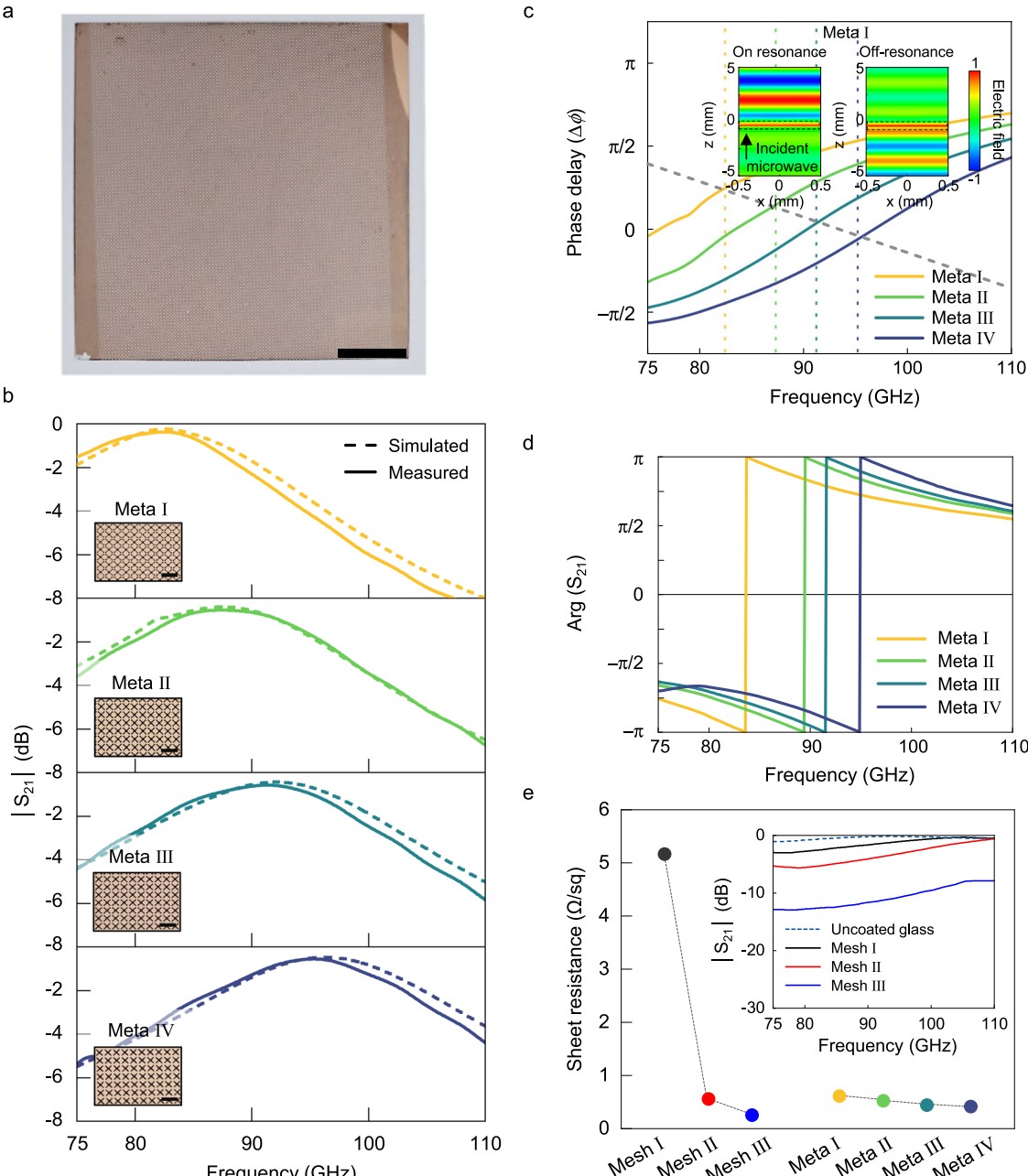

**Fig. 3 | Spectral and electric characteristics of fabricated metamaterials.**
**a** Visible camera image of a Cu metamaterial with a metal filling ratio of 71.3 %
fabricated on a glass cover. Scale bar, 2 cm. **b** Measured (solid lines) and simulated
(dashed lines) transmission ($S_{21}$) spectra (75–110 GHz) of four (Meta I–IV) fabricated
metamaterial samples. The key parameter values of each metamaterial design are
listed in Supplementary Table 3. (Insets) Visible camera image of each sample. Scale
bar, 2 mm. **c** Simulated phase delay values when an incident microwave signal is
reflected from one end of the glass substrate including the metamaterials (Meta
I–IV). The grey dashed line represents phase delay values calculated from the

analytical standing wave condition. The coloured dashed lines represent trans-
mission peak frequencies from the measured $S_{21}$ spectra of the four fabricated
metamaterial samples. (Inset) Electric field profiles of Meta I at 82.5 (on resonance)
and 110 GHz (off-resonance). **d** Measured $S_{21}$ phase values the four (Meta I–IV)
metamaterial designs. **e** Measured sheet resistance values of the fabricated samples
(Meta I–IV and Mesh I–III). (Inset) Measured transmission spectra of reference
samples based on two-dimensional mesh arrays with discrete metal filling ratios
(5 %, 30 % and 70 % for Mesh I–III, respectively). For comparison, an uncoated glass
substrate is plotted as dashed line.

are largely attributed to the asymmetric electrode configuration.
Properly designed electrodes can help minimise the temperature dif-
ferences between the centre and edges[45].

It is essential to note that transparent heaters must maintain their
pristine transparency regardless of temperature variations. To verify
this, we obtained the transmission spectra of the metamaterial-based
samples under an input voltage of 1.6 V (Supplementary Fig. 11).
Applying an input voltage of 1.6 V, corresponding to the range of
temperatures between 76–94 °C for Meta I–IV respectively, marginally

affected the transmission spectra compared to those before applying
an input voltage, thus preserving near-unity transparency at the target
frequencies. To elucidate these results, we considered two factors as
follows: (i) structural deformation by thermal expansion and (ii)
temperature-dependent electrical conductivity ($\sigma$). First, we estimated
the change in structural parameters of the metamaterials (Meta I–IV)
and obtained transmission spectra for increased temperatures ($\Delta T$) of
0 °C, 50 °C, 100 °C and 150 °C (Supplementary Fig. 12a). The simulated
results reveal that thermal expansion does not affect the transmission

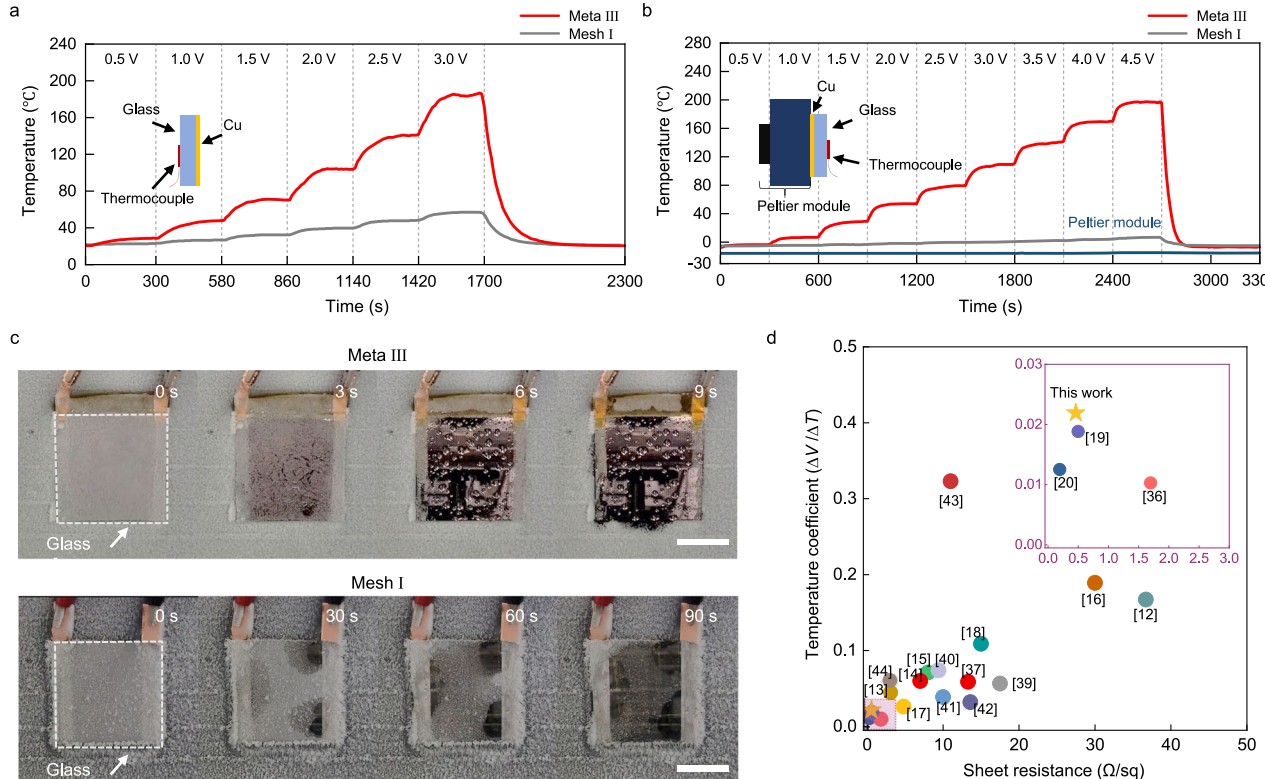

**Fig. 4 | Thermal characteristics of microwave-transparent heaters.**
**a** Temperature changes of the metamaterial-based (Meta III) and mesh-array-based (Mesh I) samples while the input voltages are varied from 0 to 3 V in increments of 0.5 V. **b** Temperature changes of the metamaterial-based (Meta III) sample while the input voltages are varied from 0 to 4.5 V in increments of 0.5 V. The samples were placed on a Peltier module set to a temperature of −20 °C. **c** Visible camera images of Meta III (top) and Mesh I (bottom) under an input voltage of 4.5 V during defrosting tests. Both samples were placed on a Peltier module set to sustain a temperature of −20 °C. Scale bar, 2.5 cm. **d** Sheet resistance and temperature coefficient values derived from previous studies. The star symbol represents our data. Details of the previous studies are described in Supplementary Table 4. (Inset) Magnified data within the highlighted area.

spectra, maintaining near-unity transmission. Secondly, we considered electrical conductivity ($\sigma$) of Cu at two discrete temperatures ($\Delta T$) of 0 °C and 100 °C, corresponding to $59.60 \times 10^6$ S/m and $45.43 \times 10^6$ S/m, respectively (Supplementary Fig. 12b). Likewise, electrical conductivity does not affect the transmittance spectra within the range of considered temperatures.

Furthermore, to evaluate the relationship between sheet resistance and heating performance, we fabricated microwave-transparent Cu metamaterial-based samples with distinct metal filling ratio (22.6, 31.6 and 71.8 % for Meta A–C, respectively) and sheet resistance values (1.21, 0.75 and 0.51 Ω/sq for Meta A–C, respectively) (Supplementary Fig. 13). Heating tests on the three fabricated samples showed the maximum temperatures of 79, 87 and 105 °C for Meta A–C, respectively at an input voltage of 2.4 V. These results indicate that lower sheet resistance correlates with enhanced heating performance. We further assessed heating capabilities of the metamaterials (Meta I–IV) under an extreme environmental condition (Fig. 4b). The samples were placed on a Peltier module set to sustain a temperature of −20 °C while monitoring temperature with input voltages from 0 to 4.5 V. While Meta III outperformed Mesh I similar with the previous result, the temperature contrast between both heaters was more pronounced. At 4.5 V, Meta III reached a temperature of 197 °C, while Mesh I achieved a temperature of merely 7 °C.

We conducted defrosting tests on both samples (Meta III and Mesh I) (Methods). Prior to the tests, the samples were placed on a Peltier module set to a temperature of −20 °C for 100 min. Subsequently, visible and thermographic camera images were captured at constant intervals of 3 s, and an input voltage of 4.5 V was applied (Fig. 4c, Supplementary Fig. 14 and Supplementary Movie 1). Similar to

the temperature measurements, we monitored the temporal change in the frozen ice layer formed on the uncoated face of the glass cover. Upon applying an input voltage, the metamaterial-based sample swiftly melted the frozen ice layer on the glass cover and completely removed the ice within 9 s. In contrast, the mesh-based sample failed to fully remove the ice layer even after 90 s. We conducted further experiments to evaluate the defrosting time as a function of input voltage (Supplementary Fig. 15). An increase in the input voltage from 0.75 – 4.5 V corresponded to a sequential decrease in defrosting time from 285 to 4 s. This result exhibits a clear trend of decreasing defrosting time with an increase in the input voltage, indicating that the use of a 12 V standard voltage will further reduce the defrosting time. However, it should be noted that the curve on the defrosting time follows an exponentially decaying nature. This elicits a message that an input voltage below the standard 12 V is sufficient for effective defrosting. Notably, the Peltier module remained active continuously during the measurements to simulate severe environmental conditions. Moreover, considering the prevalent use of a standard 12 V power supply in automotive electrical systems, the metamaterial-based heater can instantaneously eliminate any frozen ice layer in a real automotive setting. These observations suggest that the developed microwave-transparent metamaterials, featuring a high metal filling ratio, are efficient at defrosting even under demanding conditions such as deep-sub-zero temperatures.

Finally, we comparatively analysed the performance of our developed transparent heaters and those reported in previous studies[12–20,36,37,39–44] (Fig. 4d). To enable this comparison, we selected two key metrics: the temperature coefficient, which represents the change in the input voltage corresponding to the resultant

temperature change, and the sheet resistance. Noteworthily, all data for comparison were extracted from studies of visibly transparent heaters based on metal nanowires/meshes[16–20,37–39], conductive oxides[12] and their hybrid films[13–15,40–44] (Supplementary Table 4). To the best of our knowledge, our work presents the first report of microwave-transparent heaters. In terms of both temperature coefficient and sheet resistance, our device is comparable to state-of-the-art devices, primarily owing to the high metal filling of our metamaterial design.

## Discussion

We designed interconnected microwave-transparent metamaterials that operated at specific frequencies within the W band. Their maximum transmission was practically unity, excluding the absorption by the cover material, even up to a metal filling ratio of 70 %. This counterintuitive behaviour was attributed to the ultrathin metamaterial inducing an appropriate phase delay that shifted the Fabry–Perot resonance to the desired frequency. These metallic metamaterials enabled the development of transparent heaters with low sheet resistance for automotive radars, ensuring the prompt removal of thin ice layers, even at deep sub-zero temperatures.

To design microwave-transparent metamaterials, we employed a heuristic approach to maximise their metal filling ratio while anchoring their transmission peaks at specific frequencies. Machine learning based on binary optimisation (e.g. '0' and '1' for a dielectric and metal pixel, respectively) can discover two-dimensional pixelated patterns that allow for perfect microwave transmission[46–48]. In particular, the upper limit of the metal filling ratio can be expanded through iterative machine learning cycles.

Practical radar modules incorporate shielding and cover layers comprising an assembly for protection against mechanical and thermal stresses, in which a microwave-transparent heater is embedded between these components. The ability of the designed metamaterials to achieve perfect transmission was valid even within this configuration (Supplementary Fig. 16). Moreover, concealing the radar modules is crucial for maintaining the aesthetic exterior of vehicles. To address this issue, we fabricated a metamaterial composed of $SiO_2/Cr/SiO_2/Al$ (80/10/80/300 nm) layers on a black polyether ether ketone (PEEK) substrate (Supplementary Fig. 17). The fabricated structure exhibited a dark grey appearance, rendering it well-suited for discreet integration behind the emblem of vehicles. Although the developed metamaterials appear metallic to the naked eye, as shown in Fig. 3a, they establish interference-free metallic enclosures for a specific range of microwave frequencies.

Furthermore, adaptability to non-developable surfaces is crucial for broader application scope[49–51]. The developed metamaterials can sustain near-unity transmission even on a low level of curvatures because their angular spectra remain almost unchanged from 0–60° incident angles, as shown in Fig. 2c. In addition, we conducted simulations to evaluate the performance of our metamaterial on a substrate with spherical curvatures (Supplementary Fig. 18). Our metamaterial design maintained near-unity transmission on the curved substrate. Also, we fabricated metamaterial heaters using flexible substrates and obtained their $S_{21}$ spectra at various bending radii (Supplementary Fig. 19). The measured spectra exhibited that the transmission characteristics were preserved across the tested curvatures. These characteristics are also beneficial for millimetre-wave communications with various form factors, in which curtailing communication noise originating from resonance effects and managing heat dissipation are important.

## Methods

### Fabrication of microwave-transparent metamaterial heaters

Microwave-transparent metamaterials were fabricated on glass substrates (Eagle-XG) using standard semiconductor processes, including photolithography and vacuum deposition. Initially, a 2 μm-thick negative photoresist (DNR-L300-D1) was coated onto a glass substrate. The substrate was then exposed to ultraviolet radiation in a mask aligner (MA-150 e, Karl Suss) using a chrome photomask. Subsequently, Cu was deposited using an electron-beam evaporator (UEE, ULTEC). Finally, the Cu-deposited sample was lifted off by immersion in an acetone solution. The dimensions of the fabricated samples were $10 \times 10$ cm², as shown in Fig. 3a.

### Electromagnetic simulations

Commercial software (CST Studio Suite, Simulia) was employed to perform the electromagnetic simulations in the microwave frequency range (75–110 GHz). These simulations were aimed at determining the scattering parameters and phase delay values of the samples. The measured complex permittivity dispersion of the glass substrate, as shown in Supplementary Fig. 1, and Cu with a conductivity of $5.8 \times 10^7$ S/m were applied. Tetrahedral meshes were employed for simulation grids, with various scales of tetrahedrons ranging from 0.01 and 141 μm in edge lengths.

### Microwave spectrum analysis

For the transmission spectra, shown in Figs. 1c, 3b, d, e, a vector network analyser (E8362C, Agilent Technologies) with millimetre-wave VNA extenders (V10VNA2-T/R, V10VNA2-T, OML) was employed. WR-10 horn antennas were used as both the transmission and reception antennas for the microwave signals. All samples were mounted onto a sample holder fabricated using a 3D printer (310 F, Cubicon). All spectral measurements were conducted inside an anechoic microwave chamber, as shown in Supplementary Fig. 4.

### Electrical characteristics

The sheet resistances of the fabricated samples, shown in Fig. 3e, were determined using a four-point probe method with a sheet resistivity meter (MCP-T610, Mitsubishi). The measurements were conducted thrice, and the resulting average value was calculated to represent the sheet resistance of the samples.

### Thermal characteristics and defrosting experiments

The heating experiments, shown in Fig. 4a, Supplementary Fig. 9 and Supplementary Fig. 10, electrically conductive Cu tape (1181, 3 M) was attached to the two edges of the fabricated samples. An input voltage was applied to the samples through the Cu tapes using a power supply (2280 S, Keithley). The temporal evolution of temperature on the uncoated face of the glass cover was recorded using multiple adhesive thermocouples (SA1-K-72, Omega Engineering) at predetermined locations, as shown in Supplementary Fig. 10. Temperature readings were captured at 10 s intervals using a data logger (RDXL 12 SD, Omega Engineering). For the heating experiments in harsh conditions as shown in Fig. 4b, the samples were positioned on top of a Peltier cooling module (CP–200; TE Technology). An input voltage was applied to the samples through the Cu tapes using a power supply (2260B-30-72, Keithley). For the defrosting test presented in Fig. 4c and Supplementary Fig. 14, the samples were positioned on top of a Peltier cooling module (CP–200; TE Technology). The Cu-based metamaterial was in contact with the Peltier cooling module, and the uncoated face of the glass cover was exposed to air. This cooling module was connected to a temperature controller (TC-720, TE Technology) and power supply (PS-24-25, TE Technology). An input bias of 4.5 V was applied to the samples using Cu tape. Prior to the defrosting experiments, the samples were subject to an environment with a temperature of −20 °C for 140 min. Throughout the defrosting tests, visible and thermographic images of the fabricated heaters were obtained using a smartphone camera and an infrared camera (FLIR A655SC), respectively. Thermal analysis software (ResearchIR, Teledyne FLIR) was used for expedient data acquisition, image analysis, and data extraction.

## Reporting summary

Further information on research design is available in the Nature Portfolio Reporting Summary linked to this article.

## Data availability

All data that support the findings of this study are available in this article and supplementary materials. Source data provided in this paper are available at https://doi.org/10.6084/m9.figshare.25371088.

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

## Acknowledgements

S.-K.K. was supported by the National Research Foundation of Korea through the Nano Material Technology Development Program (2022M3H4A1A02046445), Basic Science Research Program (RS–2023-00207966) and the Quantum Computing Based on Quantum Advantage Challenge Research (RS–2023-00255442). E.-J.L. was supported by Graduate School Innovation office, Kyung Hee University.

## Author contributions

S.-K. K. and Y.-B. K. conceived and designed the experiments. E.-J.L. and J.-Y.K. measured the optical characteristics. E.-J.L. fabricated the structures and E.-J.L. and J.-Y.K. performed the heat transfer and electromagnetic simulations. All authors participated in the data analysis and interpretation.

## Competing interests

The authors declare no competing interests.
