## [Peer Review File · Nature Communications]

Microwave-Transparent Metallic Metamaterials for Autonomous Driving SafetyREVIEWER COMMENTS

Reviewer #1 (Remarks to the Author):

The authors mainly designed microwave-transparent metallic metamaterials. Compared with the common transparent electric heater in visible light band, the research work of transparent electric heating in microwave band is rarely reported. In addition, this work is based on the application requirements of automotive radar defogging and deicing, and also shows a good application potential. Although this paper's innovative work on microwave-transparent heating is groundbreaking and interesting, I have some concerns about whether this work can meet the high reception standards of Natural Communications:

- 1) The core innovation of this work is the design of microwave-transparent metallic metamaterials. Whether it is a milestone work in the field of electromagnetic wave design and simulation is debatable. For example, recent paper indicates that novel concepts such as optical and RF dual transparent electromagnetic devices have also been proposed (Nature Electronics, 2023, 6(7): 525-533).
- 2) The author's statement "the samples (Meta I - IV and Mesh III) featuring a metal filling ratio of approximately >65% yielded sheet resistance values of 0.41-0.61 Ω /sq, setting a record-high standard compared with transparent heaters operating in the visible regime "is not accurate enough. The sheet resistance of the metal mesh transparent electrodes has been reduced to less than 0.1 while ensuring the light transmittance of more than 80% (Advanced Materials, 2019, 31(32): 1902479, Advanced Electronic Materials, 2019, 5(5): 1800991) . The metamaterial filling rate designed by the author is relatively large. Why is the sheet resistance value relatively not high enough?
- 3) The author should use 12V automobile standard voltage for deicing and defrosting test, because the deicing speed is very important for the actual use process.
- 4) There are a lot of transparent electric heating design and manufacturing technologies on the plane substrate, even including the design of metamaterials. In the face of practical application, there are many non-developable surface substrates. Is this design and processing suitable for non-developable surface substrates
- 5) For metal mesh, increasing its filling rate will reduce its sheet resistance value, but will reduce the microwave transmittance. However, in the case of keeping the metal mesh filling rate unchanged, increasing the thickness of the metal mesh (or the aspect ratio) can also effectively reduce the metal mesh resistance, but at this time the microwave transmittance is maintained. Therefore, I wonder if metal mesh can also achieve good microwave transmittance?
- 6) If the metal electric heater is designed at the bottom or periphery of the radar, can the same electric heating effect be achieved without affecting the normal operation of the radar?

Reviewer #2 (Remarks to the Author):

In this manuscript, the authors developed a microwave-transparent, low-sheet-resistance heaters for automotive radars. The article presents a metamaterial, in which the additional phase applied to the glass cover ensures high transmittance in the W-band by meeting the first-order FP resonance condition. The transmittance of this metamaterial does not decrease as the metal filling ratio increases, thus allowing for the maintenance of a high metal filling ratio (low sheet resistance) while achieving high transmittance. The performance of the metamaterial was validated through defrosting tests. In my opinion, designing heaters using metamaterial is the main innovation of the work. While this has certain engineering applications, it does not exhibit new physics beyond the design. I suggest that it could be considered in Communication Engineering, Physical Review Applied or other specialized journals in terms of the practical structure of the idea. Several detailed suggestions are given below to make the paper more readable.

1. With temperature variation, how does the variation in the electromagnetic parameters of Cu affect the transmittance?
2. When the transmitting end of the millimeter-wave radar is in a circular shape, and the covering glass transitions from a flat surface to a curved one, how to utilize this metamaterial to achieve high transmittance on curved surfaces?
3. What will happen to the transmittance if the height of metamaterial is changed?
4. The measurement method for the complex permittivity of glass at line 96 is shown in Supplementary Fig.1. However, Supplementary Fig.1 depicts the measurement of a composite structure consisting of Cu and a glass cover, which can be confusing. the author should make necessary changes to avoid it.
5. In Supplementary Fig. 2, it can be observed that as D changes from 750um to 1000um, a peak shift towards the lower frequencies, but when D becomes 125um, the peak shifts towards the higher frequencies, showing an overall unstable trend. The increase in w_0 and w_1 causes the peak to shift towards higher frequencies, which implies a blue shift rather than a red shift.

Reviewer #3 (Remarks to the Author):

This work describes microwave-transparent, low-sheet-resistance heaters for automotive radars inspired by metamaterial design. A millimetre-thick dielectric cover acts as Fabry-Perot resonator and is covered with an electrically connected metallic layer, that is claimed to behave as a metamaterial that provides a phase delay (due to anomalous dispersion?) with near-unity transmission at microwave frequencies (75–110 GHz), while having a high metal filling ratio (> 70%). This enables microwave-transparent heaters with low sheet resistance (0.41ohm/sq) that can heat the dielectric cover with low voltage (~ 3 V).

Automotive radar is a timely application given the current interest in autonomous vehicles and driver assisted technologies. Moreover, maintaining radar functionality under adverse weather conditions is important. Transparent heaters operating in the radar operating range, i.e., at microwave frequencies, seems an important step. However, it should be placed into the context of how this problem is currently solved and how current approaches fail to address this. That is not entirely clear from the manuscript.

The manuscript explains how radar systems use a dielectric cover as a resonant cavity for microwave signals and describes this with the appropriate level of detail using eq. 1. One would expect a similar treatment to describe the operating principle of the metamaterial that is used for the approach.

However, such a treatment is missing. We only read that the layer “imparts an appropriate phase delay ... while retaining its amplitude”. The abstract mentions “anomalous dispersions of these metamaterials yield the desired phase delay”, but further elaboration is not to be found in the main text. It is thus not clear how and why the metallic layer behaves as a metamaterial and what the metamaterial properties are that the approach relies on (e.g., no effective permittivity values are given). The design seems to be inspired by an electric inductive-capacitive resonator array design (ref 28), but the geometry in the manuscript differs quite a bit from the design in ref 28. The reader would benefit greatly from a self-consistent description of the operating principle for the design, including the metamaterial properties the design relies on for its operation, before proceeding with a (heuristic) design optimization.

Angle independence is important for this application. The authors mention that “transmission at the peak frequency remained almost unchanged over incident angle of 0–60°” They say this “is consistent with the behaviour of the Fresnel equations”, but there is no further explanation given. They also write that this “indicates that the designed metamaterial is characterised by a definite permittivity value”, which is not substantiated since metamaterial properties (e.g., permittivity) are not provided.

On the thermal side, low sheet resistance is a key metric. It would be useful to relate sheet resistance to heating performance in a simple quantitative way that would enable the reader to better appreciate the thermal measurements.

Response Document

Manuscript Info

Manuscript ID: NCOMMS-23-52319-T

Title: Microwave-Transparent Metallic Metamaterials for Autonomous Driving Safety

Corresponding Author

Professor Sun-Kyung Kim (sunkim@khu.ac.kr)

Contents

Point-by-point responses to Reviewer #1	Page 2
Point-by-point responses to Reviewer #2	Page 13
Point-by-point responses to Reviewer #3	Page 22

Formatting Key

Reviewer's Comment: Blue colored font

Author Response: Black colored font

Revised Text: Red colored font

Response to Reviewer 1

General comment: The authors mainly designed microwave-transparent metallic metamaterials. Compared with the common transparent electric heater in visible light band, the research work of transparent electric heating in microwave band is rarely reported. In addition, this work is based on the application requirements of automotive radar defogging and deicing, and also shows a good application potential. Although this paper's innovative work on microwave-transparent heating is groundbreaking and interesting, I have some concerns about whether this work can meet the high reception standards of Natural Communications:

Author Response. We thank the reviewer for the thoughtful suggestions and insights. The manuscript has been rechecked, and the necessary changes have been made in accordance with the reviewer's suggestions. The responses to all comments have been prepared and given below. We look forward to working with the reviewer to move this manuscript closer to publication in this journal.

1-1. The core innovation of this work is the design of microwave-transparent metallic metamaterials. Whether it is a milestone work in the field of electromagnetic wave design and simulation is debatable. For example, recent paper indicates that novel concepts such as optical and RF dual transparent electromagnetic devices have also been proposed (Nature Electronics, 2023, 6(7): 525-533).

Author Response. Thanks to the reviewer, we have recognised the recent literature on transparent electronic devices in the microwave and visible bands. The developed devices have high transmittance values of >90% in the 18–40 GHz and 400–800 nm, which results from a nominal (~0.4%) metal filling ratio. In contrast, we, for the first time to our best knowledge, present a near-unity (~100%) (when the absorption of a glass substrate is excluded) transparency at a specific microwave frequency despite a high (~70%) metal filling ratio, thereby creating unconventional applications for autonomous driving.

To clarify this, we have revised the manuscript with the suggested literature as follows. (First paragraph, Page 2) “Despite the extensive exploration of metamaterials in previous studies, research on transmissive metamaterials in the microwave regime remains relatively limited, often constrained by low metal filling ratios⁹, and their potential applications are yet to be clarified.

Reference

9 Zu, H. R., Wu, B., Chen, B., Li, W. H., Su, T., Liu, Y. Tang, W. X., He, D. P. & Cui, T. J. Optically and radiofrequency-transparent metadevices based on quasi-one-dimensional surface plasmon polariton structures. *Nature Electronics* 6, 525-533 (2023).

1-2. The author's statement "the samples (Meta I – IV and Mesh III) featuring a metal filling ratio of approximately >65% yielded sheet resistance values of 0.41-0.61Ω/sq, setting a record-high standard compared with transparent heaters operating in the visible regime "is not accurate enough. The sheet resistance of the metal mesh transparent electrodes has been reduced to less than 0.1 while ensuring the light transmittance of more than 80% (Advanced Materials, 2019, 31(32): 1902479, Advanced Electronic Materials, 2019, 5(5): 1800991). The metamaterial filling rate designed by the author is relatively large. Why is the sheet resistance value relatively not high enough?

Author Response. As the reviewer mentioned, the literature published in Advanced Materials reported a transparent heater using a 10 μm thick Ag layer, achieving a sheet resistance value of 0.28 Ω/sq while maintaining a transmittance of 95.4% at 550 nm. The literature in Advanced Electronic Materials reported a sheet resistance of 0.03 Ω/sq and 86.5% transmittance at 550 nm from Ag nanoparticle-decorated Cu films with a thickness of 5.1 μm. We note that the difference in sheet resistance between the previous and our studies is mostly ascribed to different metal thicknesses and materials. Our microwave transparent heaters use a submicron (300 nm) Cu layer, considering the qualities of

compactness and cost-effectiveness.

To explore the practical limit of the sheet resistance of our metamaterial heaters, we have obtained the sheet resistance of an unpatterned Cu film with the same thickness of 300 nm, as shown in Fig. R1. The unpatterned Cu film exhibits the sheet resistance value of 0.09 Ω/sq . We believe that this value sets the lowest attainable sheet resistance for our metamaterial heaters. Nonetheless, we acknowledge that the sheet resistance of our metamaterial heaters can be further enhanced by employing more conductive metals (e.g., Ag and Ag-based alloys) and increasing the metal thickness.

Fig. R1. Sheet resistance of fabricated samples. Measured sheet resistance values of the fabricated samples (Meta I–IV and 300 nm-thick unpatterned Cu film). The 300 nm-thick Cu film exhibits the sheet resistance value of 0.09 Ω/sq .

To address this, we have revised the manuscript as follows. (First paragraph, Page 3) “Researchers have explored various materials for high-performance transparent heaters, such as transparent conductive oxides (TCO)¹², TCO/silver/TCO multilayers^{13–15}, silver nanowires^{16–20}, carbon-based nanomaterials^{21,22} encompassing carbon nanotubes, graphene and **microstructured metals**^{23,24}.”

Reference

- Zhu, X., Xu, Q., Li, H., Liu, M., Li, Z., Yang, K., Zhao, J., Qian, L., Peng, Z., Zhang, G., Yang, J., Wang, F., Li, D., & Lan, H. Fabrication of High-Performance Silver Mesh for Transparent Glass Heaters via Electric-Field-Driven Microscale 3D Printing and UV-Assisted Microtransfer. *Advanced Materials* 31, 1902479 (2019).
- Chen, X., Nie, S., Guo, W., Fei, F., Su, W., Gu, W., & Cui, Z. Printable High-Aspect Ratio and High-Resolution Cu Grid Flexible Transparent Conductive Film with Figure of Merit over 80 000. *Advanced Electronic Materials*, 5, 1800991 (2019)

In addition, we agree that our statement was overdressed and have revised the manuscript as follows. (First paragraph, Page 9) “Particularly, the samples (Meta I–IV and Mesh III) featuring a metal filling ratio of approximately $>65\%$ yielded sheet resistance values of 0.41–0.61 Ω/sq , **exhibiting competitive performance** compared with transparent heaters operating in the visible regime^{15–18,20,23,24,31–41}. In comparison, the sample with a nominal metal-filling ratio of 5% (Mesh I) exhibited a sheet resistance of $>5 \Omega/\text{sq}$. **Recent studies reported transparent heaters in the visible achieving sheet resistance values of 0.28 Ω/sq using a 10 μm thick Ag layer²³ and 0.03 Ω/sq from Ag nanoparticle-decorated Cu films with a thickness of 5.1 μm ²⁴. The sheet resistance of our 300 nm thick, Cu-based metamaterial heaters can be further enhanced by employing more conductive metals (e.g., Ag and Ag-based alloys) and increasing the metal thickness.”**

1-3. The author should use 12V automobile standard voltage for deicing and defrosting test, because the deicing speed is very important for the actual use process.

Author Response. As the reviewer pointed out, evaluating the thermal performance of our metamaterial heater under the 12 V automotive standard voltage should be conducted. However, as shown in Fig. R2 (also set as a new Fig. 4b), our metamaterial heater (Meta III) has already reached $\sim 200\text{ }^{\circ}\text{C}$ at an input voltage of 4.5 V while it was connected to a Peltier module set to $-20\text{ }^{\circ}\text{C}$ to emulate an extreme weather condition. In contrast, Mesh I as a reference sample reaches merely $7\text{ }^{\circ}\text{C}$ under the same condition. Please note that Mesh I is a mesh-array-based sample, featuring a metal filling ratio of 5% and microwave transmission similar to Meta III (see the results in Fig. 3e). These results show that our metamaterial heater can successfully attain sufficiently high temperatures at input voltages lower than 12 V even under harsh conditions.

Fig. R2. Thermal characteristics of microwave-transparent heater. b, Temperature changes of the metamaterial-based (Meta III) sample while the input voltages are varied from 0 to 4.5 V in increments of 0.5 V. The samples were placed on a Peltier module set to a temperature of $-20\text{ }^{\circ}\text{C}$. Inset: schematic of the measurement setup.

In addition, we have conducted new defrosting experiments on both samples (Meta III and Mesh I) at an input voltage of 4.5 V, as shown in Fig. R3 (set as a new Fig. 4c). Prior to the experiments, the samples were placed on a Peltier module set to temperature of $-20\text{ }^{\circ}\text{C}$ for 90 min. We monitored the temporal change in the frozen layer at intervals of 3 s and 30 s for Meta III and Mesh I, respectively. Upon the voltage being applied, Meta III completely removes the ice layer within 9 s. In contrast, Mesh I partially removed the ice layer even after 90 s.

Fig. R3. Thermal characteristics of microwave-transparent heater. c, Visible camera images of Meta III (top) and Mesh I (bottom) under an input voltage of 4.5 V during defrosting tests. Both samples

were placed on a Peltier module set to sustain a temperature of $-20\text{ }^{\circ}\text{C}$. Scale bar, 2.5 cm. Furthermore, we have acquired the defrosting time of our metamaterial heater as a function of input voltage. An increase in the input voltage from 0.75 to 4.5 V corresponds to a sequential decrease in deicing time from 285 to 4 s. This result exhibits a clear trend of decreasing deicing time with an increase in the input voltage, indicating that the use of a 12 V standard voltage will further reduce the deicing time. However, it should be noted that the curve on the deicing time follows an exponentially decaying function. This elicits a message that an input voltage below the standard 12 V is sufficient for effective deicing and defrosting when our metamaterial heater is employed.

Supplementary Fig. 15. Defrosting tests of the metamaterial-based transparent heater. Measured defrosting time of the metamaterial-based (Meta IV) sample while the input voltages are varied from 0.75 to 4.5 V in increments of 0.25 V. The sample was placed on a Peltier module set to sustain a temperature of $-20\text{ }^{\circ}\text{C}$.

Accordingly, we have revised the manuscript with the updated figures of Fig. 4 and Supplementary Fig. 15 as follows. (First paragraph, Page 11) “We further assessed heating capabilities of the metamaterials (Meta I–IV) under an extreme environmental condition (Fig. 4b). The samples were placed on a Peltier module set to sustain a temperature of $-20\text{ }^{\circ}\text{C}$ while monitoring temperature with input voltages from 0 to 4.5 V. While Meta III outperformed Mesh I similar with the previous result, the temperature contrast between both heaters was more pronounced. At 4.5 V, Meta III reached a temperature of $197\text{ }^{\circ}\text{C}$, while Mesh I achieved a temperature of merely $7\text{ }^{\circ}\text{C}$.”

(Second paragraph, Page 12) “Subsequently, visible and thermographic camera images were captured at constant intervals of 3 s, and an input voltage of 4.5 V was applied (Fig. 4c and Supplementary Fig. 15). ... Upon applying an input voltage, the metamaterial-based sample swiftly melted the frozen ice layer on the glass cover and completely removed the ice within 9 s. In contrast, the mesh-based sample failed to fully remove the ice layer even after 90 s. We conducted further experiments to evaluate the defrosting time as a function of input voltage (Supplementary Fig. 15). An increase in the input voltage from 0.75 to 4.5 V corresponded to a sequential decrease in defrosting time from 285 to 4 s. This result exhibits a clear trend of decreasing defrosting time with an increase in the input voltage, indicating that the use of a 12 V standard voltage will further reduce the defrosting time. However, it should be noted that the curve on the defrosting time follows an exponentially decaying nature. This elicits a message that an input voltage below the standard 12 V is sufficient for effective defrosting.”

To clarify our experiments, we have added a new supplementary file as Supplementary Movie 1. This Supplementary Movie 1 includes the snapshots of Fig. 4c and Supplementary Fig. 14 in the visible and infrared region, respectively.

Fig. 4. Thermal characteristics of microwave-transparent heater. **a**, Temperature changes of the metamaterial-based (Meta III) and mesh-array-based (Mesh I) samples while the input voltages are varied from 0 to 3.0 V in increments of 0.5 V. **b**, Temperature changes of the metamaterial-based (Meta III) sample while the input voltages are varied from 0 to 4.5 V in increments of 0.5 V. The samples were placed on a Peltier module set to sustain a temperature of -20°C . **c**, Visible camera images of Meta III (top) and Mesh I (bottom) under an input voltage of 4.5 V during defrosting tests. Both samples were placed on a Peltier module set to sustain a temperature of -20°C . Scale bar, 2.5 cm. **d**, Sheet resistance and temperature coefficient values derived from previous studies. The star symbol represents our data. Details of the previous studies are described in Table S4. (Inset) Magnified data within the highlighted area.

1-4. There are a lot of transparent electric heating design and manufacturing technologies on the plane substrate, even including the design of metamaterials. In the face of practical application, there are many non-developable surface substrates. Is this design and processing suitable for non-developable surface substrates.

Author Response. As the reviewer pointed out, adaptability to non-developable surfaces is crucial in the face of practical applications. Researchers have explored various approaches to design curved metamaterials via segmentation or transformation optics⁴⁸⁻⁵⁰.

Reference

- 48 Jang, Y., Yoo, M., & Lim, S. Conformal metamaterial absorber for curved surface. *Optics Express* 21, 24163 (2013).
- 49 Zhang, C., Yang, J., Cao, W., Yuan, W., Ke, J., Yang, L., Cheng, Q., & Cui, T. Transparently curved metamaterial with broadband millimeter wave absorption. *Photonics Research* 7, 478 (2019).
- 50 Pendry, J. B., Schurig, D., & Smith, D. R. Controlling Electromagnetic Fields. *Science* 312, 1780-1782 (2006)

Our findings specifically target radar heaters for automobiles. In this context, radar systems are predominantly mounted on relatively flat surfaces, such as the front bonnet/grill and rear bumper²⁶.

These surfaces, albeit not perfectly planar, exhibit a low level of curvature that can be approximated to a plane. We speculate that our metamaterial design can be effective on these low-curvature surfaces, supported by the angular S_{21} spectra in the Fig. 2c; this result shows near-unity transmittance of our metamaterials from 0–60° incidence angles.

Reference

26 Rishabh, B. et al. Managing heating element operational parameters, WO 2022197956A1 (2022)

In addition, we have conducted simulations to evaluate the performance of our metamaterial on substrates with non-developable curvatures. As shown in Supplementary Fig. 18, we modelled a unit metamaterial pattern on a simple spherical substrate, characterised by a radius of curvatures (R) of ∞ (i.e., planar), 25, 10, 5 and 2.5 mm, respectively. The results present that the S_{21} spectra from $R = 25$ to 5 mm remain nearly unchanged, although the overall spectrum shifts slightly toward lower frequencies at $R = 2.5$ mm. We speculate that this spectral shift is attributed to structural deformations resulting from the stretching of the periphery of the metamaterial, thus effectively increasing the parameter D , as illustrated in Supplementary Fig. 2b. Note that, even at $R = 2.5$ mm, the transmission level is almost preserved, maintaining the Fabry-Perot resonance induced transparency. While our current focus has been on low-curvature surfaces in automotive radar systems, we will be committed to extending our research to more complex surface geometries.

Supplementary Fig. 18. Metamaterial on a spherical curvature. **a**, Schematic illustration of a unit metamaterial pattern on a spherically curved substrate characterised by radius of curvature R . The structural parameters of the metamaterial (a , D , w_0 , w_1 and g) are identical to the Meta III in the manuscript. **b**, Simulated S_{21} spectra as a function of the radius of curvature R . The black line is the S_{21} spectra from the metamaterials on a planar substrate, which is the same data displayed in Fig. 3b.

Furthermore, we have fabricated metamaterial heaters using self-healing Au-Ga liquid metals and flexible polymer substrates (Fig. R4), which will be more suitable for non-planar applications. The measured S_{21} spectra show that both samples achieve high transmission levels of >-1.0 dB.

Fig. R4. Adaptation of metamaterials to curved surface substrate. **a**, Visible camera image of metamaterial (Meta I) using Au-Ga liquid metal. **b**, Measured transmission (S_{21}) spectra of Cu-based metamaterial and Au-Ga liquid metal-based metamaterial. **c**, Visible camera image of metamaterial applied to a polycarbonate-based substrate. **d**, Measured transmission (S_{21}) spectra of metamaterial (Meta I) on the glass substrate and flexible polymer substrate.

To further support our response of applicability to non-planar surfaces, we have obtained the S_{21} spectra of four fabricated metamaterial heaters (Meta I–IV) on polymer at different bending radii (R) (Supplementary Fig. 19). We selected $R = 5.07$ and 3.18 cm to emulate the geometry of automobiles, which correspond to the central and peripheral curvatures of a commercial automobile’s front bumper. The measured S_{21} spectra exhibit that the transmission characteristics are preserved across the tested curvatures, reinforcing the practical applicability of our design to automotive radar systems with non-planar surfaces.

Supplementary Fig. 19. Transmission measurement of metamaterials on curved flexible substrates. **a**, Schematic illustration of transmission measurement setup to apply defined curvature on a flexible polymer substrate. **b**, Measured transmission (S_{21}) spectra of four fabricated metamaterials (Meta I–IV) on polymer substrates. Solid lines indicate S_{21} spectra on planar surfaces for comparison.

To address this, we have revised the manuscript with new supplementary figures as follows. (Third paragraph, Conclusion) “Although the developed metamaterials appear metallic to the naked eye, as shown in Fig. 3a, they establish interference-free metallic enclosures for a specific range of microwave frequencies. Furthermore, adaptability to non-developable surfaces is crucial for broader application scope⁴⁸⁻⁵⁰. The developed metamaterials can sustain near-unity transmission even on a low level of curvatures because their angular spectra remain almost unchanged from 0–60° incident angles, as shown in Fig. 2c. In addition, we conducted simulations to evaluate the performance of our metamaterial on a substrate with spherical curvatures (Supplementary Fig. 18). Our metamaterial design maintained near-unity transmission on the curved substrate. Also, we fabricated metamaterial heaters using flexible substrates and obtained their S_{21} spectra at various bending radii (Supplementary Fig. 19). The measured spectra exhibited that the transmission characteristics were preserved across the tested curvatures.”

1-5. For metal mesh, increasing its filling rate will reduce its sheet resistance value, but will reduce the microwave transmittance. However, in the case of keeping the metal mesh filling rate unchanged, increasing the thickness of the metal mesh (or the aspect ratio) can also effectively reduce the metal mesh resistance, but at this time the microwave transmittance is maintained. Therefore, I wonder if metal mesh can also achieve good microwave transmittance?

Author Response. We concur with the reviewer that increasing the metal thickness can effectively reduce sheet resistance, similar to the effect of increasing the metal filling ratio. However, such structural adjustments can also affect the spectral response, particularly microwave transparency. To investigate the effect of metal thickness on transmission spectra, we have conducted simulations in which the thicknesses (h) of mesh and metamaterial arrays incrementally increased (Supplementary Fig. 7). For both structures, increasing h up to 5 μm does not significantly change the transmission spectra

compared to that at $h = 0.3 \mu\text{m}$. When h exceeds $10 \mu\text{m}$, the transmission spectra of the mesh arrays show noticeable degradation within the W band. These results suggest that adjusting meshes to match the sheet resistance of our metamaterials with high metal filling ratios by increasing their metal thickness poses challenges in maintaining transmission levels.

Supplementary Fig. 7. Transmission (S_{21}) spectra as a function of metal thickness. Simulated W band transmission spectra of (a) mesh and (b) metamaterial arrays with the thickness (h) of 0.1, 0.3, 0.5, 1, 5, 10, 50 and 100 μm , respectively. Metal filling ratios are 5, 30 and 70% for Mesh I–III and 71.3, 68.3, 66.3 and 64.3% for Meta I–IV. Spectra for $h = 0.3 \mu\text{m}$ are the same data as displayed in Fig. 3b.

More importantly, as demonstrated in Fig. 2e, the metal filling ratio affects not only sheet resistance but also temperature uniformity. To facilitate a direct comparison, we have obtained the temperature distribution for Cu-based mesh and metamaterial arrays, using a heat transfer simulation (Fig. R5). The simulated results show that both structures achieve more uniform temperature distributions as the metal filling ratio increases. However, as discussed in the manuscript, the mesh array with a 70% metal filling ratio (Mesh III) exhibited transmission levels ranging from -13.0 dB to -7.8 dB within the W band, whereas the metamaterial array with a 71.3% metal filling ratio (Meta I) sustains near-unity transmission levels (> -1.0 dB) at the target microwave frequency. Comparison between the mesh array with a 5% metal filling ratio (Mesh I), which has a decent level of transmission, and the metamaterial array with a 71.3% metal filling ratio (Meta I) reveals a significant difference in temperature uniformity. Hence, our metamaterial design takes advantage of enhanced transparency and thermal uniformity simultaneously.

Fig. R5. Heat transfer simulation on mesh and metamaterial arrays. A, b, Simulated temperature distribution on a glass cover incorporating (a) mesh arrays and (b) metamaterial arrays. Metal filling ratios are 5%, 30% and 70% (Mesh I–III) for mesh arrays and 22.6% and 71.3% (Meta I–IV) for metamaterial arrays. The variation in colour within the area corresponds to the relative temperature gradients, with white representing the highest temperature and darker colour indicating a decrease in temperature.

In response to the reviewer’s comment, we have revised the manuscript with the new supplementary figures. (First paragraph, Page 9) “**Although increasing the metal thickness of Mesh I to tens of microns can effectively reduce sheet resistance, this approach degrades transmission within the W band (Supplementary Fig. 7) and leads to nonuniform temperature distribution due to the low metal filling ratio of the mesh array (Supplementary Fig. 8).**”

Supplementary Fig. 8. Heat transfer simulation on mesh arrays. Simulated temperature distribution on a glass cover (Eagle-XG) incorporating mesh arrays. Metal filling ratios are 5%, 30% and 70% (Mesh I–III) for mesh arrays. The variation in colour within the area corresponds to the relative temperature gradients, with white representing the highest temperature and darker colour indicating a decrease in temperature.

1-6. If the metal electric heater is designed at the bottom or periphery of the radar, can the same electric heating effect be achieved without affecting the normal operation of the radar?

Author Response. We recognise that there might have been a misconception arising from our initial schematic representation in Fig. 1b; it seemed that a heater and radar sensor were in direct contact. In a practical scenario, a shielding layer/film exists between the two elements. Overall, a radar system consists of a radar chip containing antennas and sensors, enclosed by shielding and cover layers for protection against mechanical and thermal stresses. To address this, we have revised the schematic illustration in Fig. 1b.

Fig. 1. Concept of microwave-transparent heaters for automotive radars. **b**, Schematics illustrating a radar sensor module with and without a heater in adverse weather conditions. A thin layer of frost or ice accumulated on the cover of the module considerably reflects or scatters both incoming and outgoing signals.

Accordingly, we considered this assembly configuration in which a microwave-transparent heater was embedded between the protective layers (Supplementary Fig. 16). The simulated results show that near-perfect transmission at a target microwave frequency is achieved within this configuration. To clarify this, we have also updated the schematic in Supplementary Fig. 16a.

Supplementary Fig. 16. Microwave-transparent heaters with a protective cover and substrate. **a**, Schematic of a metamaterial embedded glass substrate, illustrating its structural parameters of a , D , w_0 , w_1 , h and g values. **b**, Simulated transmission (S_{21}) spectra (75–110 GHz) of metamaterial embedded glass substrates while w_0 and g values are varied. For all the simulations, $a = 1000 \mu\text{m}$, $D = 1200 \mu\text{m}$, $w_1 = 350 \mu\text{m}$ and $h = 0.3 \mu\text{m}$. Their metal (Cu) filling ratios are within 40–70%. These simulations support near-unity transmission of embedded metamaterial designs with high metal filling ratios.

To address this, we have revised the manuscript as follows. (Third paragraph, Conclusion) “Practical radar modules incorporate **shielding and cover layers comprising an assembly for protection against mechanical and thermal stresses, in which a microwave-transparent heater is embedded between these components.**”

Response to Reviewer 2

In this manuscript, the authors developed a microwave-transparent, low-sheet-resistance heaters for automotive radars. The article presents a metamaterial, in which the additional phase applied to the glass cover ensures high transmittance in the W-band by meeting the first-order FP resonance condition. The transmittance of this metamaterial does not decrease as the metal filling ratio increases, thus allowing for the maintenance of a high metal filling ratio (low sheet resistance) while achieving high transmittance. The performance of the metamaterial was validated through defrosting tests. In my opinion, designing heaters using metamaterial is the main innovation of the work. While this has certain engineering applications, it does not exhibit new physics beyond the design. I suggest that it could be considered in Communication Engineering, Physical Review Applied or other specialized journals in terms of the practical structure of the idea. Several detailed suggestions are given below to make the paper more readable.

Author Response. We thank the reviewer for these thoughtful suggestions and insights. The manuscript has been rechecked, and the necessary changes have been made in accordance with the reviewer's suggestions. The responses to all comments have been prepared and given below. We look forward to working with the reviewer to move this manuscript closer to publication in this journal.

2-1. With temperature variation, how does the variation in the electromagnetic parameters of Cu affect the transmittance?

Author Response. To investigate the effects of temperature on our Cu-based metamaterials, we have obtained their transmission spectra under an input voltage of 1.6 V (Supplementary Fig. 11). Applying an input voltage of 1.6 V, corresponding to the range of temperatures between 76–94 °C for Meta I–IV, marginally affects the transmission spectra compared to those before applying an input voltage, thus preserving near-unity transparency at the target frequencies.

Supplementary Fig. 11. Transmission (S_{21}) spectra of metamaterials with temperature variations. Measured transmission (S_{21}) spectra of the fabricated samples (Meta I–IV) at input voltages of 0 and 1.6 V, corresponding to the range of temperatures between 76–94 °C for Meta I–IV respectively.

To elucidate these results, we consider two factors as follows: (i) structural deformation by thermal expansion and (ii) temperature-dependent electrical conductivity (σ). First, we use the thermal expansion formula,

$$\Delta L = \alpha L_0 \Delta T,$$

where the thermal expansion coefficient (α) of Cu is $1.7 \times 10^{-5} \text{ K}^{-1}$. Then, we have estimated the change in structural parameters (D , w_0 , w_1 and g) of the metamaterials (Meta I–IV) and obtained transmission spectra for increased temperatures (ΔT) of 0 °C, 50 °C, 100 °C and 150 °C (Supplementary Fig. 12). The simulated results show that thermal expansion does not affect the transmittance spectra, maintaining near-unity transmission. Second, we consider electrical conductivity (σ) of Cu at two discrete temperatures (ΔT) of 0 and 100 °C, corresponding to $59.60 \times 10^6 \text{ S/m}$ and $45.43 \times 10^6 \text{ S/m}$, respectively. Likewise, electrical conductivity does not affect transmittance spectra within the range of considered temperatures.

Supplementary Fig. 12. S_{21} spectra of the metamaterials at different temperatures. **a**, Simulated microwave transmittance (S_{21}) spectra of the metamaterials (Meta I–IV), by adjusting their structural parameters (D , w_0 , w_1 and g) in consideration of temperature increases (ΔT) at 0 °C, 50 °C, 100 °C and 150 °C. **b**, Simulated microwave transmittance (S_{21}) spectra of the metamaterials (Meta I–IV) with the electric conductivity of $59.60 \times 10^6 \text{ S/m}$ and $45.43 \times 10^6 \text{ S/m}$, respectively.

To address this, we have revised the manuscript with new supplementary figures as follows. (First paragraph, Page 10) “It is essential to note that transparent heaters must maintain their pristine transparency regardless of temperature variations. To verify this, we obtained the transmission spectra of the metamaterial-based samples under an input voltage of 1.6 V (Supplementary Fig. 11). Applying an input voltage of 1.6 V, corresponding to the range of temperatures between 76–94 °C for Meta I–IV respectively, marginally affected the transmission spectra compared to those before applying an input voltage, thus preserving near-unity transparency at the target frequencies. To elucidate these results, we considered two factors as follows: (i) structural deformation by thermal expansion and (ii) temperature-dependent electrical conductivity (σ). First, we estimated the change in structural parameters of the metamaterials (Meta I–IV) and obtained transmission spectra for increased temperatures (ΔT) of 0 °C, 50 °C, 100 °C and 150 °C (Supplementary Fig. 12a). The simulated results reveal that thermal expansion does not affect the transmission spectra, maintaining near-unity transmission. Secondly, we considered electrical conductivity (σ) of Cu at two discrete temperatures (ΔT) of 0 °C and 100 °C, corresponding to $59.60 \times 10^6 \text{ S/m}$ and $45.43 \times 10^6 \text{ S/m}$, respectively (Supplementary Fig. 12b). Likewise, electrical conductivity does not affect the transmittance spectra within the range of considered temperatures.”

2-2. When the transmitting end of the millimeter-wave radar is in a circular shape, and the covering glass transitions from a flat surface to a curved one, how to utilize this metamaterial to achieve high transmittance on curved surfaces?

Author Response. As the reviewer pointed out, adaptability to non-developable surfaces is crucial in the face of practical applications. Researchers have explored various approaches to design curved metamaterials via segmentation or transformation optics⁴⁸⁻⁵⁰.

Reference

- 48 Jang, Y., Yoo, M., & Lim, S. Conformal metamaterial absorber for curved surface. *Optics Express* 21, 24163 (2013).
- 49 Zhang, C., Yang, J., Cao, W., Yuan, W., Ke, J., Yang, L., Cheng, Q., & Cui, T. Transparently curved metamaterial with broadband millimeter wave absorption. *Photonics Research* 7, 478 (2019).
- 50 Pendry, J. B., Schurig, D., & Smith, D. R. Controlling Electromagnetic Fields. *Science* 312, 1780-1782 (2006)

Our findings specifically target radar heaters for automobiles. In this context, radar systems are predominantly mounted on relatively flat surfaces, such as the front bonnet/grill and rear bumper²⁶. These surfaces, albeit not perfectly planar, exhibit a low level of curvature that can be approximated to a plane. We speculate that our metamaterial design can be effective on these low-curvature surfaces, supported by the angular S_{21} spectra in the Fig. 2c; this result shows near-unity transmittance of our metamaterials from 0–60° incidence angles.

Reference

- 26 Rishabh, B. et al. Managing heating element operational parameters, WO 2022197956A1 (2022)

In addition, we have conducted simulations to evaluate the performance of our metamaterial on substrates with non-developable curvatures. As shown in Supplementary Fig. 18, we modelled a unit metamaterial pattern on a simple spherical substrate, characterised by a radius of curvatures (R) of ∞ (i.e., planar), 25, 10, 5 and 2.5 mm, respectively. The results present that the S_{21} spectra from $R = 25$ to 5 mm remain nearly unchanged, although the overall spectrum shifts slightly toward lower frequencies at $R = 2.5$ mm. We speculate that this spectral shift is attributed to structural deformations resulting from the stretching of the periphery of the metamaterial, thus effectively increasing the parameter D , as illustrated in Supplementary Fig. 2b. Note that, even at $R = 2.5$ mm, the transmission level is almost preserved, maintaining the Fabry-Perot resonance induced transparency. While our current focus has been on low-curvature surfaces in automotive radar systems, we will be committed to extending our research to more complex surface geometries.

Supplementary Fig. 18. Metamaterial on a spherical curvature. **a**, Schematic illustration of a unit metamaterial pattern on a spherically curved substrate characterised by radius of curvature R . The structural parameters of the metamaterial (a , D , w_0 , w_1 and g) are identical to the Meta III in the manuscript. **b**, Simulated S_{21} spectra as a function of the radius of curvature R . The black line is the S_{21} spectra from the metamaterials on a planar substrate, which is the same data displayed in Fig. 3b.

Furthermore, we have fabricated metamaterial heaters using self-healing Au-Ga liquid metals and flexible polymer substrates (Fig. R4), which will be more suitable for non-planar applications. The measured S_{21} spectra show that both samples achieve high transmission levels of >-1.0 dB.

Fig. R4. Adaptation of metamaterials to curved surface substrate. **a**, Visible camera image of metamaterial (Meta I) using Au-Ga liquid metal. **b**, Measured transmission (S_{21}) spectra of Cu-based metamaterial and Au-Ga liquid metal-based metamaterial. **c**, Visible camera image of metamaterial applied to a polycarbonate-based substrate. **d**, Measured transmission (S_{21}) spectra of metamaterial (Meta I) on the glass substrate and flexible polymer substrate.

To further support our response of applicability to non-planar surfaces, we have obtained the S_{21} spectra of four fabricated metamaterial heaters (Meta I–IV) on polymer at different bending radii (R) (Supplementary Fig. 19). We selected $R = 5.07$ and 3.18 cm to emulate the geometry of automobiles, which correspond to the central and peripheral curvatures of a commercial automobile’s front bumper. The measured S_{21} spectra exhibit that the transmission characteristics are preserved across the tested curvatures, reinforcing the practical applicability of our design to automotive radar systems with non-planar surfaces.

Supplementary Fig. 19. Transmission measurement of metamaterials on curved flexible substrates. **a**, Schematic illustration of transmission measurement setup to apply defined curvature on a flexible polymer substrate. **b**, Measured transmission (S_{21}) spectra of four fabricated metamaterials (Meta I–IV) on polymer substrates. Solid lines indicate S_{21} spectra on planar surfaces for comparison.

To address this, we have revised the manuscript with new supplementary figures as follows. (Third paragraph, Conclusion) “Although the developed metamaterials appear metallic to the naked eye, as shown in Fig. 3a, they establish interference-free metallic enclosures for a specific range of microwave frequencies. Furthermore, adaptability to non-developable surfaces is crucial for broader application scope⁴⁸⁻⁵⁰. The developed metamaterials can sustain near-unity transmission even on a low level of curvatures because their angular spectra remain almost unchanged from 0–60° incident angles, as shown in Fig. 2c. In addition, we conducted simulations to evaluate the performance of our metamaterial on a substrate with spherical curvatures (Supplementary Fig. 18). Our metamaterial design maintained near-unity transmission on the curved substrate. Also, we fabricated metamaterial heaters using flexible substrates and obtained their S_{21} spectra at various bending radii (Supplementary Fig. 19). The measured spectra exhibited that the transmission characteristics were preserved across the tested curvatures.”

2-3. What will happen to the transmittance if the height of metamaterial is changed?

Author Response. To investigate the effect of metal thickness on transmission spectra, we have conducted simulations in which the thicknesses (h) of mesh and metamaterial arrays incrementally increased (Supplementary Fig. 7). For both structures, increasing h up to 5 μm does not significantly change the transmission spectra compared to that at $h = 0.3 \mu\text{m}$. When h exceeds 10 μm , the transmission spectra of the mesh arrays show noticeable degradation within the W band. These results suggest that adjusting meshes to match the sheet resistance of our metamaterials with high metal filling ratios by increasing their metal thickness poses challenges in maintaining transmission levels.

Supplementary Fig. 7. Transmission (S_{21}) spectra as a function of metal thickness. Simulated W band transmission spectra of (a) mesh and (b) metamaterial arrays with the thickness (h) of 0.1, 0.3, 0.5, 1, 5, 10, 50 and 100 μm , respectively. Metal filling ratios are 5, 30 and 70% for Mesh I–III and 71.3, 68.3, 66.3 and 64.3% for Meta I–IV. Spectra for $h = 0.3 \mu\text{m}$ are the same data as displayed in Fig. 3b.

More importantly, as demonstrated in Fig. 2e, the metal filling ratio affects not only sheet resistance but also temperature uniformity. To facilitate a direct comparison, we have obtained the temperature distribution for Cu-based mesh and metamaterial arrays, using a heat transfer simulation (Fig. R5). The simulated results show that both structures achieve more uniform temperature distributions as the metal filling ratio increases. However, as discussed in the manuscript, the mesh array with a 70% metal filling ratio (Mesh III) exhibited transmission levels ranging from -13.0 dB to -7.8 dB within the W band, whereas the metamaterial array with a 71.3% metal filling ratio (Meta I) sustains near-unity transmission levels (> -1.0 dB) at the target microwave frequency. Comparison between the mesh array with a 5% metal filling ratio (Mesh I), which has a decent level of transmission, and the metamaterial array with a 71.3% metal filling ratio (Meta I) reveals a significant difference in temperature uniformity. Hence, our metamaterial design takes advantage of enhanced transparency and thermal uniformity simultaneously.

Fig. R5. Heat transfer simulation on mesh and metamaterial arrays. A, b, Simulated temperature distribution on a glass cover incorporating (a) mesh arrays and (b) metamaterial arrays. Metal filling ratios are 5%, 30% and 70% (Mesh I–III) for mesh arrays and 22.6% and 71.3% (Meta I–IV) for metamaterial arrays. The variation in colour within the area corresponds to the relative temperature gradients, with white representing the highest temperature and darker colour indicating a decrease in temperature.

In response to the reviewer’s comment, we have revised the manuscript with the new supplementary figures as follows. (First paragraph, Page 9) “**Although increasing the metal thickness of Mesh I to tens of microns can effectively reduce sheet resistance, this approach degrades transmission within the W band (Supplementary Fig. 7) and leads to nonuniform temperature distribution due to the low metal filling ratio of the mesh array (Supplementary Fig. 8).**”

Supplementary Fig. 8. Heat transfer simulation on mesh arrays. Simulated temperature distribution on a glass cover (Eagle-XG) incorporating mesh arrays. Metal filling ratios are 5%, 30% and 70% (Mesh I–III) for mesh arrays. The variation in colour within the area corresponds to the relative temperature gradients, with white representing the highest temperature and darker colour indicating a decrease in temperature.

2-4. The measurement method for the complex permittivity of glass at line 96 is shown in Supplementary Fig.1. However, Supplementary Fig.1 depicts the measurement of a composite structure consisting of Cu and a glass cover, which can be confusing. the author should make necessary changes to avoid it.

Author Response. We apologise for this confusion. To address this, we have revised the configuration in the schematic of Supplementary Fig. 1a.

Supplementary Fig. 1. Complex permittivity dispersion of a glass cover. a, Schematic of a measurement setup for obtaining complex permittivity dispersion at microwave frequencies. The permittivity dispersion of a sample is determined through a retrieval process with transmitted and reflected signals.

2-5. In Supplementary Fig. 2, it can be observed that as D changes from 750 μm to 1000 μm , a peak shift towards the lower frequencies, but when D becomes 1250 μm , the peak shifts towards the higher frequencies, showing an overall unstable trend. The increase in w_0 and w_1 causes the peak to shift towards higher frequencies, which implies a blue shift rather than a red shift.

Author Response. As the reviewer correctly pointed out, the peak shifts towards lower frequencies with increasing D until the topology of the metamaterial remains unchanged. As shown in Fig. R6, an increase in D up to 1000 μm induces a peak shift towards lower frequencies. However, the transmission peak shifts towards higher frequencies when D exceeds 1000 μm due to improper definition of the inductive loop. To avoid this confusion, we have limited the adjustment of the structural parameter D to 1000 μm . Increasing D from 600 to 1000 μm results in a shift of the peak towards lower frequencies.

Fig. R6. Primary structural parameter D of a metamaterial design. a, Schematics of metamaterials with variations in the structural parameter D at 750, 1000 and 1250 μm . For all schematics, a , w_0 , w_1 and g are fixed at 1000, 50, 50 and 30 μm , respectively. **b,** Simulated S_{21} spectra with an increase in the structural parameter of D at 600, 800 and 1000 μm . For all schematics, a , w_0 , w_1 and g are fixed at 1000, 50, 50 and 30 μm , respectively.

In addition, we have opted the terms 'towards lower frequencies' and 'towards higher frequencies' instead of using the terms 'redshift' and 'blueshift'. Accordingly, we have revised the manuscript with an updated supplementary Fig. 2b as follows: (Second paragraph, page 5) "In this geometry, increasing the values of D , w_0 and w_1 individually enhanced the metal filling ratio. However, in terms of spectral tuning, an increase in D induced a lower frequency shift of transmission peaks, which is attributed to an increase in the length of inductive loops³². Conversely, an increase in w_0 and w_1 resulted in a peak shift towards higher frequencies because this structural change effectively reduced the loop inductance (Supplementary Fig. 2 and Supplementary Fig. 3)."

Reference

32 Schurig, D., Mock, J. J. & Smith, D. R. Electric-field-coupled resonators for negative permittivity metamaterials. *Applied Physics Letters* 88, 041109 (2006).

Supplementary Fig. 2. Primary structural parameters of a metamaterial design. **a**, Schematic illustrating the unit cell of a metamaterial array. For all the simulations, a and h are fixed at 1000 and 0.3 μm , respectively. **b–e**, Simulated S_{21} spectra with individually increasing the parameters of **(b)** D , **(c)** w_0 , **(d)** w_1 , **(e)** g , while the other parameters are fixed.

Response to Reviewer 3

General comment: This work describes microwave-transparent, low-sheet-resistance heaters for automotive radars inspired by metamaterial design.

A millimetre-thick dielectric cover acts as Fabry-Perot resonator and is covered with an electrically connected metallic layer, that is claimed to behave as a metamaterial that provides a phase delay (due to anomalous dispersion?) with near-unity transmission at microwave frequencies (75–110 GHz), while having a high metal filling ratio (> 70%). This enables microwave-transparent heaters with low sheet resistance (0.41 ohm/sq) that can heat the dielectric cover with low voltage (~3 V).

Author Response. We thank the reviewer for the thoughtful suggestions and insights. The manuscript has been rechecked, and the necessary changes have been made in accordance with the reviewer's suggestions. The responses to all comments have been prepared and given below. We look forward to working with the reviewer to move this manuscript closer to publication in this journal.

3-1. Automotive radar is a timely application given the current interest in autonomous vehicles and driver assisted technologies. Moreover, maintaining radar functionality under adverse weather conditions is important. Transparent heaters operating in the radar operating range, i.e., at microwave frequencies, seems an important step. However, it should be placed into the context of how this problem is currently solved and how current approaches fail to address this. That is not entirely clear from the manuscript.

Author Response. As the reviewer pointed out, maintaining stable radar functionality under adverse weather conditions is essential. Currently, one-dimensional (1D) wire grid arrays have been attempted for microwave-transparent heaters²⁶. However, they must be designed with low metal filling ratios to maintain high microwave transmission⁹.

Reference

26 Rishabh, B. et al. Managing heating element operational parameters, WO 2022197956A1 (2022).

To clarify this, we have revised the manuscript as follows. (First paragraph, Page 3) “Furthermore, few attempts have been made to develop transparent heaters that operate in the microwave regime. A feasible solution for microwave-transparent heaters could be a one-dimensional (1D) metallic wire array²⁶, akin to the wire-grid polarisers used in display applications²⁷. This design can perfectly transmit microwaves exclusively for a specific polarisation. However, any alteration in the polarisation of microwaves reflected from the surroundings undermines their detectability²⁸. More importantly, the high transmission of a 1D metallic wire array for a specific polarisation vanishes as the metallic filling ratio increases, which is a general trend for hyperbolic metamaterials²⁹.”

To investigate the dependence of metal filling ratio on microwave transmission in a 1D wire array, we have conducted simulations with various widths (w) ranging from 100 to 700 μm (Supplemental Fig. S6). When the width (w) reaches 700 μm , corresponding to a metal filling ratio of 70%, the transmission levels at 77 GHz are -2.1 dB (62% transmittance) for TM polarisation and -23.3 dB (0.5% transmittance) for TE polarisation. These results suggest that 1D wire arrays are limited in simultaneously achieving microwave transparency and high metal filling ratios.

Supplementary Fig. 6. The microwave-transmittance (S_{21}) spectra of 1D wire grid structure. a, Schematic illustrating 1D wire grid arrays, characterised by structural parameters of p , w and h . For all simulations, p and h are fixed at $1000 \mu\text{m}$ and $0.3 \mu\text{m}$, respectively. **b, c,** Simulated S_{21} spectra with increasing the parameter of w for **(b)** transverse magnetic (TM) polarisation and **(c)** transverse electric (TE) polarisation, while the other parameters are fixed.

To address this, we have revised the manuscript with a new supplementary figure as follows. (First paragraph, Page 9) “At the metal filling ratio of 70%, the level of transmission was between -13.0 and -7.8 dB within the W band. In addition, we simulated transmittance spectra of 1D metallic wire arrays with different metal filling ratios (10%, 30%, 50% and 70%) for two orthogonal polarisations (Supplementary Fig. 6). At a metal filling ratio of 70%, the highest levels of transmission in the W band for transverse magnetic and electric polarisations were -1.9 and -19.7 dB, respectively. The 1D metallic wire arrays exhibited a gradual decrease in transmission when the metal filling ratios increased, as observed in the mesh arrays.”

3-2. The manuscript explains how radar systems use a dielectric cover as a resonant cavity for microwave signals and describes this with the appropriate level of detail using eq. 1. One would expect a similar treatment to describe the operating principle of the metamaterial that is used for the approach. However, such a treatment is missing. We only read that the layer “imparts an appropriate phase delay ... while retaining its amplitude”. The abstract mentions “anomalous dispersions of these metamaterials yield the desired phase delay”, but further elaboration is not to be found in the main text. It is thus not clear how and why the metallic layer behaves as a metamaterial and what the metamaterial properties are that the approach relies on (e.g., no effective permittivity values are given).

Author Response. As the reviewer pointed out, obtaining an effective permittivity would offer a clear explanation of the behaviour of the metallic layer as a metamaterial. Extracting the effective permittivity requires that the considered metamaterial layer has a deep-subwavelength thickness in free space [R1]. However, the inductive-capacitive resonators within our metallic layer are directly coupled with the Fabry-Perot resonance of the millimetre-thick glass substrate. This coupling made it challenging to derive the effective permittivity of the metallic layer from the measured complex transmission (S_{21}) and reflection (S_{22}) coefficients.

Reference

R1 X. Liu, L. Gan, B. Yang, Millimeter-wave free-space dielectric characterization, *Measurement*, 179, 109472 (2021).

We acknowledge that Fig. 3d and its discussion did not sufficiently clarify how the metallic layer induces the desired phase delay to shift its resonant frequency. To address this, we have conducted simulations wherein the metallic layer is placed on a semi-infinite glass substrate to negate multiple reflections (Fig. R7a). In this configuration, a change in the reflection coefficient (S_{22}) represents the degree of phase delay when an incident microwave signal is reflected from one end of the glass substrate

including the metallic layer. For comparison, we have calculated the degree of phase delay ($\Delta\phi_s$) required to shift the Fabry-Perot resonance to the target frequency, using the analytical standing wave condition as follows:

$$\Delta\phi_s = 2\Delta kt = 2 \frac{2\pi(f_{\text{FP}} - f_t)}{c} nt$$

where f_{FP} is the Fabry-Perot resonance frequency of the glass substrate (at 93.4 GHz), f_t is the target frequency, c is the speed of light, and n and t are the refractive index and thickness of the glass substrate. As shown in Fig. R7b, the simulated reflection phase delay values for the four different metallic layers (Meta I–IV) support the analytical standing wave conditions at 82.2, 87.4, 91.4 and 95.6 GHz, respectively. These frequencies precisely match the transmission peak frequencies observed in the measured S_{21} spectra. Despite their deep-subwavelength thickness (approximately 0.1% of the wavelength), the metallic layers effectively induce significant phase changes, behaving as metamaterials with an anomalously high effective permittivity.

Fig. R7. Phase compensation induced by metamaterials. **a**, Schematic illustration of a single-path propagation simulation to obtain reflection phase delay of a metamaterial. **b**, Simulated phase delay values when an incident microwave signal is reflected from one end of the glass substrate including the metamaterials (Meta I–IV). The grey dashed line represents phase delay values calculated from the analytical standing wave condition. The coloured dashed lines represent transmission peak frequencies from the measured S_{21} spectra of the four fabricated metamaterial samples.

In response to the reviewer, we have revised the manuscript with the updated Fig. 4c as follows. (First paragraph, Page 8) “To verify the phase compensation of the metamaterials, we calculated the difference in the phase delay ($\Delta\phi$) that occurred when a microwave signal is reflected from one end of the glass substrate including the metamaterials (Meta I–IV) (Fig. 3c). The simulated $\Delta\phi$ values for the four different metamaterials support the analytical standing wave conditions at 82.2, 87.4, 91.4 and 95.6 GHz, respectively. These frequencies precisely match the transmission peak frequencies observed in the measured S_{21} spectra from Fig 3b. Despite their deep-subwavelength thickness (approximately 0.1% of the wavelength), the metallic surfaces effectively induce significant phase changes, behaving as metamaterials with an anomalously high effective permittivity.” We have also replaced the equation (1) as follows.

$$2 \frac{2\pi f}{c} nt + \phi_1 + \phi_2 = 2m\pi$$

Fig. 3. Spectral and electric characteristics of fabricated metamaterials. c, Simulated phase delay values when an incident microwave signal is reflected from one end of the glass substrate including the metamaterials (Meta I–IV). The grey dashed line represents phase delay values calculated from the analytical standing wave condition. The coloured dashed lines represent transmission peak frequencies from the measured S_{21} spectra of the four fabricated metamaterial samples.

3-3. The design seems to be inspired by an electric inductive-capacitive resonator array design (ref 28), but the geometry in the manuscript differs quite a bit from the design in ref 28. The reader would benefit greatly from a self-consistent description of the operating principle for the design, including the metamaterial properties the design relies on for its operation, before proceeding with a (heuristic) design optimization.

Author Response. To address this, we have updated our manuscript with a more relevant reference (Ref. 28) to reflect our design inspiration (Second paragraph, page 5) “Our metamaterial design was initiated from a recognised electric inductive-capacitive resonator array²⁸ to which we added interconnection bars between adjacent unit cells to establish electrical connectivity (Fig. 2a).”

Reference

- 28 Padilla, W. J., Aronsson, M. T., Highstrete, C., Lee, M., Taylor, A. J. & Averitt, R. D. Electrically resonant terahertz metamaterials: Theoretical and experimental investigations. *Physical Review B* 75, 041102(R) (2007).

In addition, to elucidate the operating principles of our design, we have conducted simulations to obtain electromagnetic field profiles at the resonant frequency of our metamaterial design (Meta III) (Supplementary Figure 3). The simulated results reveal where capacitive and inductive phenomena occur, thus providing a clear visualisation of the impact of structural parameters on the capacitive-inductive resonance. Accordingly, we have revised the manuscript with a new supplementary figure as follows. (Second paragraph, page 5) “In this geometry, increasing the values of D , w_0 and w_1 individually enhanced the metal filling ratio. However, in terms of spectral tuning, an increase in D induced a lower frequency shift of transmission peaks, which is attributed to an increase in the length of inductive loops³². Conversely, an increase in w_0 and w_1 resulted in a peak shift toward higher frequencies, as this effectively reduced the loop inductance (Supplementary Fig. 2 and Supplementary Fig. 3). Although g (the separation between curved units) moderately impacted the metal filling ratio, it significantly tuned the overall spectrum with minimal alteration through the modulation of capacitive coupling.”

Reference

- 32 Schurig, D., Mock, J. J. & Smith, D. R. Electric-field-coupled resonators for negative permittivity metamaterials. *Applied Physics Letters* 88, 041109 (2006).

Supplementary Fig. 3. Electromagnetic response of the metamaterials. **a**, Electric- and **b**, magnetic-field intensity profiles at the cross-section of the Meta III, in response to an x-polarised microwave signal at 91.3 GHz.

3-4. Angle independence is important for this application. The authors mention that “transmission at the peak frequency remained almost unchanged over incident angle of 0–60°” They say this “is consistent with the behaviour of the Fresnel equations”, but there is no further explanation given. They also write that this “indicates that the designed metamaterial is characterised by a definite permittivity value”, which is not substantiated since metamaterial properties (e.g., permittivity) are not provided.

Author Response. We apologise for this confusion. As stated in the response #3-2, our metamaterial is incorporated on a glass substrate, which is engineered to maximise transmittance at a specific frequency through a phase delay. To avoid this confusion, we have removed the terms permittivity and Fresnel equations. (Second paragraph, page 6) “The level of transmission at the peak frequency remained almost unchanged over incident angle of 0–60° (Fig. 2c)., which is consistent with the behaviour of the Fresnel equations and indicates that the designed metamaterial is characterised by a definite permittivity value. Given the numerical aperture of automotive radars, angular tolerance is important²⁹.”

(abstract, page 1) “~~The anomalous dispersions of~~ These metamaterials yield the desired phase delay to adjust Fabry–Perot resonance at each target frequency.”

3-5. On the thermal side, low sheet resistance is a key metric. It would be useful to relate sheet resistance to heating performance in a simple quantitative way that would enable the reader to better appreciate the thermal measurements.

Author Response. As the reviewer pointed out, we have recognised the importance of relating sheet resistance to heating performance. To address this, we have fabricated new metamaterial samples (Meta A and Meta B) with sheet resistance values (i.e., lower metal filling ratios) higher than those of the samples presented in the manuscript (Meta I–IV) (Supplementary Fig. 13). For comparison, we have fabricated a new reference sample corresponding to the design of Meta I, which is referred to as Meta C. The measured sheet resistances for Meta A–C are 1.21, 0.75 and 0.51 Ω/sq , respectively. To evaluate the heating performance of the fabricated samples, temperatures were monitored at their centre under an input voltage of 2.4 V. The Meta A–C reach temperatures of 79.3, 86.7 and 105.3 $^{\circ}\text{C}$, respectively, indicating that lower sheet resistance correlates with enhanced heating performance.

Supplementary Fig. 13. Heating tests of metamaterial-based samples with various metal filling ratios. **a**, Design schematics of Meta A–C with metal filling ratio of 22.6%, 31.6% and 71.8%, respectively. The key parameters of the fabricated metamaterials are listed in Table S1. $h = 0.4 \mu\text{m}$ for these samples. **b**, Measured sheet resistance values of the fabricated samples on glass substrates. **c**, Temporal changes in temperature of the fabricated samples under an input voltage of 2.4 V.

To address this, we have revised the manuscript with a new supplementary figure as follows. (Second paragraph, page 11) “Furthermore, to evaluate the relationship between sheet resistance and heating performance, we fabricated microwave-transparent Cu metamaterial-based samples with distinct metal filling ratio (22.6, 31.6 and 71.8% for Meta A–C, respectively) and sheet resistance values (1.21, 0.75 and 0.51 Ω/sq for Meta A–C, respectively) (Supplementary Fig. 13). Heating tests on the three fabricated samples showed the maximum temperatures of 79, 87 and 105 $^{\circ}\text{C}$ for Meta A–C, respectively at an input voltage of 2.4 V. These results indicate that lower sheet resistance correlates with enhanced heating performance.”

REVIEWER COMMENTS

Reviewer #1 (Remarks to the Author):

Although the author explained the my suggestion and question in detail, I also think that this kind of metamaterial design has a good application potential for automotive radar defogging and deicing. But I'll stick with the original point, which is that while the design is good, I think it's far-fetched to meet NC Publishing standards. Although the author has demonstrated its potential for small curvature surfaces, I think that if the author can demonstrate that this design is suitable for large curvature surfaces and can be manufactured on non-developable large curvature surfaces, i think this may be a breakthrough, is enough to be published in NC. All in all, the current results are far-fetched, hope the author to make further in-depth revision and improvement.

Reviewer #2 (Remarks to the Author):

Thank you for the authors' responses and revisions. Following the revisions, the authors addressed the queries related to the design. However, these responses did not introduce any new innovations on the original. The consistent maintenance of a high transmittance has been a fundamental benchmark since the advent of metamaterials. There are numerous designs available for achieving high transmittance using metamaterials [PRL 113, 023902 (2014), Nat. Comm. 12, 4523 (2021)], and even better results can be achieved. However, the design featured in this manuscript does not present any novel advancements in this aspect. Therefore, the novelty of this work is not enough and I do not recommend this work to be published on Nature Communications as its high specification.

Reviewer #3 (Remarks to the Author):

The authors have addressed my comments.

Response Document

Manuscript Info

Manuscript ID: NCOMMS-23-52319-B

Title: Microwave-Transparent Metallic Metamaterials for Autonomous Driving Safety

Corresponding Author

Professor Sun-Kyung Kim (sunkim@khu.ac.kr)

Contents

Point-by-point responses to Reviewer #1	Page 2
Point-by-point responses to Reviewer #2	Page 3

Formatting Key

Reviewer's Comment: Blue colored font

Author Response: Black colored font

Revised Text: Red colored font

Response to Reviewer 1

1-1. Although the author explained the my suggestion and question in detail, I also think that this kind of metamaterial design has a good application potential for automotive radar defogging and deicing. But I'll stick with the original point, which is that while the design is good, I think it's far-fetched to meet NC Publishing standards. Although the author has demonstrated its potential for small curvature surfaces, I think that if the author can demonstrate that this design is suitable for large curvature surfaces and can be manufactured on non-developable large curvature surfaces, I think this may be a breakthrough, is enough to be published in NC. All in all, the current results are far-fetched, hope the author to make further in-depth revision and improvement.

Author Response. We appreciate your suggestion regarding further demonstration on larger non-developable curvatures because such demonstration can extend the use of our metamaterial design to practical applications other than automotive radars. However, we believe that fulfilling this request will be certainly beyond the scope of this study, given that i) our metamaterial design specifically targets 'low curvature' radar heaters for automobiles and ii) extending our research to more complex geometries is completely a new task, requiring sophisticated transformation optics. In the current manuscript, we demonstrated through both simulations and experiments that the perfect transmission of our metamaterial design was preserved across the tested curvatures that correspond to the geometries of a commercial automobile's bonnet.

Nevertheless, we have conducted a preliminary test to validate whether our metamaterial sample is mechanically robust to complex geometries such as a saddle-shaped curvature, as shown in Fig. R1. Given that our metamaterial design maintains perfect transmission even at high incident angles up to 60° , as shown in Fig. 2c, we anticipate that its transmission characteristics will remain almost the same on such a complex geometry.

Fig. R1. Adaptability of our metamaterial to complex surfaces. Visible camera image of a Meta I sample with a saddle-shaped curvature.

Response to Reviewer 2

2-1. Thank you for the authors' responses and revisions. Following the revisions, the authors addressed the queries related to the design. However, these responses did not introduce any new innovations on the original. The consistent maintenance of a high transmittance has been a fundamental benchmark since the advent of metamaterials. There are numerous designs available for achieving high transmittance using metamaterials [PRL 113, 023902 (2014), Nat. Comm. 12, 4523 (2021)], and even better results can be achieved. However, the design featured in this manuscript does not present any novel advancements in this aspect. Therefore, the novelty of this work is not enough and I do not recommend this work to be published on Nature Communications as its high specification.

Author Response. We thank the reviewer for suggesting references relevant to our study. However, the literature published in Physical Review Letters reported a 50% transmittance with a polarisation extinction ratio of 20:1 at 'optical' frequencies (e.g., 1.5 μm wavelength) using tri-layer metamaterials. Each layer comprised patterned submicron gold sheets at metal filling ratios of ~34%. The other literature published in Nature Communications reported antireflective and wavefront-controllable metamaterials for microwave 'cloaking' applications. These metamaterials consisted of patterned metallic sheets with metal filling ratios of ~35%. In contrast, our study reports metamaterials, featuring a closed looped (i.e., electrically interconnected) design with a high metal filling ratio (>70%), provide perfect transmission at a specific frequency within the W-band and prompt deicing performance even at deep sub-zero temperatures.

In the revised manuscript, we have added one of the suggested literatures as a reference to the relevant sentence. (First paragraph, Page 5) "These metamaterials adjust the standing wave condition to the desired microwave frequency according to the degree of phase delay, thus enabling perfect transmission at that frequency³¹.

Reference

- 31 Chu, H., et al. Invisible surfaces enabled by the coalescence of anti-reflection and wavefront controllability in ultrathin metasurfaces. *Nature Communications* **12**, 4523 (2021).